# SCOOBDOOB: SCHRÖDINGER BRIDGE WITH DOOB'S $h$-TRANSFORM FOR MOLECULAR DYNAMICS

## ABSTRACT

The slow processes of stochastic dynamical systems can be captured by Molecular Dynamics (MD) simulations, which approximate transition matrices describing how probabilities evolve over metastable conformations. Standard approaches such as Markov State Models (MSMs) extract dominant conformations and transition statistics via eigendecomposition, but face scalability and generalization limits. Here, we introduce **Sch**rö**d**inger **B**ridge with **Doob**'s $h$-Transform (**ScooBDoob**), a discrete bridge-matching framework that models metastable dynamics by tilting MSM transition rates through Doob's transform to generate optimal stochastic paths between prescribed initial and terminal ensembles. We show that ScooBDoob preserves spectral stability of slow modes during training, recovers rare transition pathways with density-aware regularization, and generalizes zero-shot across temperatures. Experiments on the Müller-Brown potential and the Aib9 peptide demonstrate accurate kinetics and robust endpoint-conditioned rollouts, highlighting broad applicability to biomolecular dynamics.

## 1 INTRODUCTION

Simulating molecular dynamics (MD) trajectories accurately and efficiently remains a fundamental challenge in computational chemistry, particularly when predicting rare transition events between metastable states (Lewis et al., 2025a). Such events are crucial for understanding biological processes like protein folding, ligand binding, and conformational dynamics, but occur over long timescales, making direct computational simulations prohibitively expensive (Ghosh and Ranjan, 2020; Vincoff et al., 2025). Markov State Models (MSMs) have emerged as a popular approach for approximating these slow processes by representing continuous trajectories as discrete microstates and modeling transitions between these states as Markovian jumps (Chodera and Noé, 2014; Trubiano and Hagan, 2024; Pande et al., 2010). By deriving transition probability matrices from MD data, MSMs efficiently summarize long-term dynamical behavior, significantly reducing computational complexity and enabling more tractable analysis of complex biomolecular systems (Chodera and Noé, 2014; Trubiano and Hagan, 2024; Pande et al., 2010).

However, MSMs face substantial challenges in practice. First, eigendecomposition of transition matrices is a crucial step for extracting dynamical information, but can lead to numerical instability and inaccuracies if eigenvectors are unconstrained (Frank et al., 2022). Unstable eigenvectors can produce physically unrealistic predictions, which undermines the reliability of MSMs for critical biological applications. Furthermore, MD simulation data is inherently sparse in regions of conformational space that correspond to rare transitions, resulting in poorly estimated transition probabilities and limited predictive accuracy (Konovalov et al., 2021; Frank et al., 2022). Sparse data render MSM-derived trajectories highly sensitive to sampling variability and noise, thereby limiting their generalizability to unseen conformations and conditions.

Recent methods have leveraged generative models to sample transition paths between metastable states by framing trajectory generation as a stochastic control problem. These approaches include optimization of the Onsager-Machlup (OM) functional or a related control Lagrangian to produce high-likelihood paths under learned dynamics (Raja et al., 2025; Du et al., 2024), and diffusion-based samplers trained with off-policy learning to efficiently approximate the transition path distribution (Seong et al., 2025; Holdijk et al., 2023). More broadly, Schrödinger bridge formulations (Liu et al., 2023) provide a principled framework for path sampling under endpoint constraints, and recent

advances in stochastic optimal control further connect bridge problems to tractable learning objectives (Liu et al., 2025). Collectively, these methods demonstrate the promise of conditioning generative dynamics on endpoint constraints to study rare events, bypassing the need for collective variables or retraining on system-specific data.

In this work, we introduce **Sch**rödinger **B**ridge with **Doob**'s $h$-Transform (**ScooBDoob**), a novel Schrödinger bridge formulation explicitly designed to enhance the robustness, stability, and generalization capabilities of MSM-based methods. Our framework integrates three key advancements:

1. **Parameterization of the Doob-Tilted Transition Matrix.** To condition the transition path on a target meta-stable state, we leverage Doob's $h$-transform to tilt the unconditional MSM transition matrix and train our parameterized model to match the optimal Schrödinger bridge. This enables the efficient simulation of feasible transition paths despite energy barriers.

2. **Density-Aware Regularization.** We introduce density-aware reweighting, which adjusts transition probabilities based on empirical MD sampling density, significantly enhancing robustness against data sparsity and sampling variability.

3. **Stiefel-Constrained Eigenvector Optimization.** We explicitly constrain eigenvectors to the Stiefel manifold, ensuring numerical stability and physically meaningful directional transitions, thus addressing the instability associated with unconstrained eigendecompositions.

We provide a detailed discussion on related works in Appendix A.

## 2 PRELIMINARIES

### 2.1 LEARNING DISCRETE SCHRÖDINGER BRIDGES

**Schrödinger Bridge Problem** The Schrödinger Bridge (SB) problem aims to find the *optimal* probability path measure $\mathbb{P}$ from samples of an initial distribution $\boldsymbol{x}_0 \sim \mu_0$ to samples from a final distribution $\boldsymbol{x}_N \sim \mu_N$. The optimal solution is defined as the path measure $\mathbb{P}^{\text{SB}}$ with marginals $\mu_0$ and $\mu_n$ that minimizes the KL-divergence to a reference path measure $\mathbb{Q}$

$$\mathbb{P}^{\text{SB}} = \min_{\mathbb{P}}\{\text{KL}(\mathbb{P}\|\mathbb{Q}) : \mathbb{P}_0 = \mu_0, \mathbb{P}_N = \mu_N\} \tag{1}$$

where $\mathbb{Q}$ can be defined as standard Brownian motion in continuous state spaces and a Dirichlet process in discrete state spaces. Note that $\mathbb{P}^{\text{SB}} \neq \mathbb{Q}$ as $\mathbb{P}$, since it must satisfy the boundary constraints $\mathbb{P} = \mu_0$ and $\mathbb{P}_T = \mu_T$.

**Continuous-Time Markov Chains** In discrete state spaces $\mathcal{X} = \{1, \dots, m\}$, time-varying stochastic process $(\boldsymbol{X}_t)_{t\in[0,T]}$ over the time horizon $[0, T]$ is considered a **continuous-time Markov chain (CTMC)** if it can be characterized by a transition rate matrix or *generator* $\boldsymbol{Q}_t \in \mathbb{R}^{\mathcal{X}\times\mathcal{X}}$ defined as

$$\boldsymbol{Q}_t(x, y) = \lim_{\Delta t \to 0} \frac{1}{\Delta t}\big[\mathbb{P}(\boldsymbol{X}_{t+\Delta t} = y|\boldsymbol{X}_t = x) - \boldsymbol{1}_{x=y}\big] \tag{2}$$

where $\mathbb{P}(\boldsymbol{X}_{t+\Delta t} = y|\boldsymbol{X}_t = x)$ is the probability of making a discrete "jump" from state $x$ at time $t$ to state $y$ at time $t + \Delta t$ and $\boldsymbol{1}_{x=y}$ is an indicator function that equals 1 if $x = y$. By taking the limit as $\Delta t$, the generator defines the *instantaneous* jump probability at time $t$. By definition, all entries of the generator are non-negative for $x \neq y$ (i.e., $\boldsymbol{Q}_t(x, y) \geq 0$) and the diagonal entries are defined as $\boldsymbol{Q}_t(x, x) = -\sum_{y\neq x}\boldsymbol{Q}_t(x, y)$.

**Doob's $h$-Transform** The Doob's $h$-transform is a theoretically-grounded method to condition a transition rate matrix (or *generator*) $\boldsymbol{Q}_t(x, y) \in \mathbb{R}^{m\times m}$ of a CTMC $(\boldsymbol{X}_t)_{t\in[0,T]}$ to a target state $z$ at time $T$ by *tilting* it via the conditional probability function $h_t(\boldsymbol{x})$.

$$\boldsymbol{Q}_t(x, y; z) = \boldsymbol{Q}_t(x, y)\frac{\mathbb{P}(\boldsymbol{X}_T = z|\boldsymbol{X}_t = y)}{\mathbb{P}(\boldsymbol{X}_T = z|\boldsymbol{X}_t = x)} - \delta_{xy}\sum_u \boldsymbol{Q}_t(x, u)\frac{\mathbb{P}(\boldsymbol{X}_T = z|\boldsymbol{X}_t = u)}{\mathbb{P}(\boldsymbol{X}_T = z|\boldsymbol{X}_t = x)} \tag{3}$$

where $\mathbb{P}(\boldsymbol{X}_T = z|\boldsymbol{X}_t = y)$ is the conditional probability of transitioning to a state $z$ at time $t$ given the current state $\boldsymbol{X}_t = x$ and $\delta_{xy}$ is the Dirac delta function that returns 1 when $x = y$ and 0 otherwise. Intuitively, this transform decreases the transition rate $x \to y$ if the probability of transitioning to $z$ from state $x$ is higher than from state $y$ and increases the transition rate if the probability of transitioning to $z$ from state $y$ is higher than from state $x$.

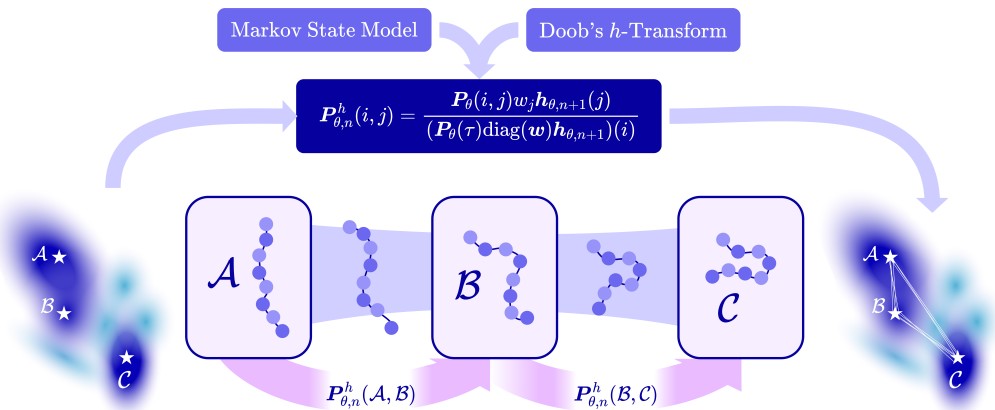

Figure 1: **Schrödinger Bridge with Doob's $h$-Transform (ScooBDoob).** ScooBDoob models stochastic transition paths between metastable states by learning Markov State Models (MSMs) of molecular dynamics trajectories and conditioning on target end-states using Doob's $h$-Transform.

## 2.2 MARKOV STATE MODELS

**Molecular Dynamics** A molecular dynamics (MD) simulation produces a time-ordered trajectory of molecular conformations, represented as Cartesian positions at discrete timesteps (Scherer et al., 2015). For larger biomolecules, the dimensionality of the MD features grows prohibitively large, resulting in computational bottlenecks when simulating their trajectories. Coarse-graining techniques have aimed to lower the dimensionality of MD features by finding *collective variables (CVs)* that largely capture the degrees of freedom of a molecule's conformation over time (Ingólfsson et al., 2014; Joshi and Deshmukh, 2021).

**Time-lagged Independent Component Analysis (TICA)** Time-lagged independent component analysis (TICA; Pérez-Hernández et al. (2013)) is a method for reducing the high-dimensional feature space of molecular systems to a set of Collective Variables (CVs) that determine the primary degrees of freedom responsible for the slow transitions in MD simulations. Consider an MD snapshot of a $d$-dimensional molecular system at time $t$ as $\boldsymbol{x}(t) \in \mathbb{R}^d$. Then, the time-lagged covariance matrices are defined as

$$\text{Cov}_{00} = \mathbb{E}\big[\boldsymbol{x}(t)\boldsymbol{x}(t)^\top\big], \quad \text{Cov}_{0\tau} = \mathbb{E}\big[\boldsymbol{x}(t)\,\boldsymbol{x}(t+\tau)^\top\big] \tag{4}$$

where the expectation is over the trajectory frames and $\tau$ is the chosen lag time. To determine the CVs, TICA solves the generalized eigenproblem

$$\text{Cov}_{0\tau}\,\boldsymbol{u}_i = \lambda_i\,\text{Cov}_{00}\boldsymbol{u}_i, \qquad i \in \{1, \dots, d\} \tag{5}$$

where $\boldsymbol{u}_i \in \mathbb{R}^d$ are the eigenvectors and $\lambda_i$ are the corresponding eigenvalues, and $t = -\tau/\ln \lambda_i$. With the top $k$ eigenvalues sorted as $1 = |\lambda_1| \geq |\lambda_2| \geq \cdots \geq |\lambda_d|$, we construct a projection matrix $\boldsymbol{U} \in \mathbb{R}^{n \times k}$ with columns being the corresponding eigenvectors, which projects $\boldsymbol{x}(t) \in \mathbb{R}^d$ to a $k$-dimensional feature vector $\boldsymbol{y}(t)$ as

$$\boldsymbol{y}(t) = \boldsymbol{U}^\top \boldsymbol{x}(t), \quad \boldsymbol{U} = [\boldsymbol{u}_1, \dots, \boldsymbol{u}_k] \tag{6}$$

**Constructing the Markov State Model** We can cluster the TICA-projected $\boldsymbol{y}(t)$ into $m$ discrete microstates and represent state transitions over a lag time $\tau$ by a **Markov State Model (MSM)**. Formally, an MSM at lag time $\tau$ is defined as a stochastic matrix $\boldsymbol{P}(\tau) \in \mathbb{R}^{m \times m}$ matrix of transition probabilities:

$$\boldsymbol{P}(\tau) \in \mathbb{R}^{m \times m}, \quad \boldsymbol{P}_{ij}(\tau) = \mathbb{P}(\boldsymbol{X}_{t+\tau} = j \mid \boldsymbol{X}_t = i) = \frac{C(i, j; \tau)}{\sum_{j'} C(i, j'; \tau)}, \tag{7}$$

where $C(i, j; \tau)$ is the observed count of transitions from state $i$ at time $t$ to state $j$ at time $t+\tau$. The construction of an MSM has a natural connection to CTMCs, where the transition probabilities define a stochastic trajectory between discrete micro-states, which motivates our work.

---

**Algorithm 1 ScooBDoob**: Schrödinger Bridge with Doob's $h$-Transform

---

1: **Input:** Observed count of transitions $i \to j$ at $\tau$ lag $C(i, j; \tau)$ for all $i, j \in \{1, \ldots, m\}$
2: **while** Training **do**
3:    $\boldsymbol{P}_{ij}(\tau) \leftarrow \frac{C(i,j;\tau)}{\sum_{j'} C(i,j';\tau)}, \quad \boldsymbol{P}(\tau) \leftarrow [\boldsymbol{P}_{ij}(\tau)]$
4:    $\boldsymbol{V}(i) \leftarrow \alpha/(C_i + 1), \quad \boldsymbol{w}(i) \leftarrow \exp(-\tau \boldsymbol{V}(i))$            $\triangleright$ *density-aware weights*
5:    $\boldsymbol{h}_N^V \leftarrow \nu$                                $\triangleright$ *initialize terminal condition*
6:    **for** $n$ in $N - 1, \ldots, 0$ **do**
7:      $\boldsymbol{h}_n^V \leftarrow \boldsymbol{P}(\tau)(\text{diag}(\boldsymbol{w})\boldsymbol{h}_{n+1}^V)$           $\triangleright$ *compute tilted distributions*
8:      $\boldsymbol{P}_n^V(i, j) \leftarrow \frac{\boldsymbol{P}_{ij}(\tau)\boldsymbol{w}(j)\boldsymbol{h}_{n+1}^V(j)}{(\boldsymbol{P}(\tau)\text{diag}(\boldsymbol{w})\boldsymbol{h}_{n+1}^V)(i)}, \quad \boldsymbol{P}_n^V \leftarrow [\boldsymbol{P}_n^V(i, j)]$
9:
10:    **end for**
11:    **for** micro-state $i$ in $1, \ldots, m$ **do**            $\triangleright$ *train generator for each state $i$*
12:      $\boldsymbol{P}_{n,\theta}^h(i, \cdot) \leftarrow \text{NN}(\theta)$
13:      Compute loss $\mathcal{L}_{\text{total}}(\theta) = \mathcal{L}_{\text{MSM}}(\theta) + \gamma_{\text{bridge}}\mathcal{L}_{\text{bridge}}(\theta) + \gamma_{\text{stief}}\mathcal{L}_{\text{stief}}(\theta)$
14:      Optimize $\theta$ with $\nabla_\theta \mathcal{L}_{\text{total}}$
15:    **end for**
16: **end while**
17: **return** parameterized transition predictor $\boldsymbol{P}_\theta(\tau) : [0, 1] \to \mathbb{R}^{m \times m}$

---

# 3 SCOOBDOOB: SCHRÖDINGER BRIDGE WITH DOOB'S $h$-TRANSFORM

We introduce **Schrödinger Bridge with Doob's $h$-Transform** (**ScooBDoob**), a discrete Schrödinger bridge framework that learns stochastic transitions between metastable states in molecular systems using a Doob-transformed Markov State Model (MSM). ScooBDoob is capable of modelling discrete transition probabilities between MD microstates without requiring knowledge of the underlying potential energy landscape, enabling flexible generalization to molecular systems without known energies and sparse MD data.

## 3.1 PROBLEM SETUP

While MD is critical for exploring conformational landscapes and reaction pathways of biomolecular systems, the performance is hindered by two prominent challenges. First, MD requires **well-defined and transferable force fields** that accurately capture intermolecular and intramolecular forces (Kaminski and Jorgensen, 1996; Zhu et al., 2012). While classical force fields enable fast simulations, they rely on several assumptions that limit the expressivity of the simulation to model rare or heterogeneous phenomena. ML-based force fields increase expressivity (Arts et al., 2023; Charron et al., 2025; Lewis et al., 2025b); however, they are biased towards the interactions seen in the training data and remain limited in their ability to generalize to unseen systems.

Second, many crucial processes, such as protein folding and allosteric switches, occur between multiple low-energy, *meta-stable states*, where transitions away from the state are rare and occur over long time-scales (Noé and Clementi, 2017). This makes these **rare processes prohibitively expensive to simulate**, especially for larger systems. Techniques that aim to coerce these transitions over smaller timescales (Ensing et al., 2006; Branduardi et al., 2012; Bussi and Branduardi, 2015; Ghosh and Ranjan, 2020) often undermine the probabilistic nature of these transitions and miss intermediate states.

These challenges motivate the development of **data-centric approaches for learning MD trajectories** (Jing et al., 2024; Daigavane et al.; Lu et al., 2025; Tan et al., 2025; Rehman et al., 2025; Wang et al., 2024) that bypass the need for defined energy landscapes and can generate feasible maps between meta-stable states that align with the data manifold, while accounting for the sparsity of MD data. Notably, **ScooBDoob** addresses *all* of these challenges by **(1)** learning probabilistic transition rates *directly* from MD trajectory data, bypassing the need for external force-fields, **(2)** conditioning discrete transitions on target states grounded in Doob's $h$-Transform theory, and **(3)** amplifying regions of low data density with Feynman-Kac reweighting.

## 3.2 DEFINING ENDPOINT-CONDITIONED TRANSITIONS BETWEEN META-STABLE STATES

**Doob's $h$-Transform for Target-Conditioned Transition Rates**   Given an unconditional transition matrix $\boldsymbol{P}(\tau) \in \mathbb{R}^{m \times m}$, we can steer trajectories toward a terminal distribution $\nu \in \Delta^{m-1}$ over $T = N\tau$ steps by recursively define the distribution at each step $\boldsymbol{h}_n \in \Delta^{m-1}$ backward in time.

$$\boldsymbol{h}_N = \nu, \quad \boldsymbol{h}_n = \boldsymbol{P}(\tau)\boldsymbol{h}_{n+1}, \quad n \in \{N-1, \ldots, 0\}. \tag{8}$$

Then, we construct the time-dependent Doob-conditioned transition matrix as

$$\boldsymbol{P}_n^h(i,j) = \boldsymbol{P}_{ij}(\tau)\frac{\boldsymbol{h}_{n+1}(j)}{\boldsymbol{h}_n(i)}, \quad \sum_{j=1}^m \boldsymbol{P}_n^h(i,j) = 1. \tag{9}$$

**Density-Aware Regularization via Feynman-Kac for Sparse MD Data**   To mitigate bias toward over-sampled basins and encourage coverage of sparsely visited regions, we introduce a density-aware non-negative potential for each micro-state $\boldsymbol{V} : \{1, \ldots m\} \to \mathbb{R}_{\geq 0}$ proportional to the empirical outgoing transition density.

$$C_i = \sum_{j \neq i} C_{ij}(\tau), \quad \rho(i) = \frac{\max(C_i, 1)}{\bar{C}}, \quad \boldsymbol{V}(i) = \alpha\rho(i), \tag{10}$$

where $C_{ij}(\tau)$ is the number of observed MD transitions at lag $\tau$ from state $i$ to state $j$ and $C_i$ denotes the total number of outgoing transitions from state $i$. $\bar{C}$ is the mean of $C_i$. $\alpha \geq 0$ is a hyperparameter controlling the penalization of sparsely sampled transitions. With the potential, we define a weight vector $\boldsymbol{w} \in \mathbb{R}^m$ containing the weights of each microstate $w_j = \exp(-\tau\boldsymbol{V}(j))$.

Given a time horizon of $T = N\tau$ with target distribution $\nu \in \Delta^{m-1}$, we define the target-conditioned density-aware probability distributions $\boldsymbol{h}_n^V \in \Delta^{m-1}$ at each time increment from $n \in \{N, N-1, \ldots, 0\}$ as

$$\boldsymbol{h}_N^V = \nu, \quad \boldsymbol{h}_n^V = \boldsymbol{P}(\tau)\left(\text{diag}(\boldsymbol{w})\boldsymbol{h}_{n+1}^V\right) \tag{11}$$

where $\text{diag}(\boldsymbol{w})\boldsymbol{h}_{n+1}^V$ reweights the probability of each state at time $n+1$ by its corresponding density weight $w_j$, thereby encouraging the likelihood of transitioning into sparsely sampled intermediate states. The resulting density-aware Doob kernel at each time step $n \in \{1, \ldots, N\}$ is defined as

$$\boldsymbol{P}_n^h(i,j) = \frac{\boldsymbol{P}_{ij}(\tau) \, w_j \, \boldsymbol{h}_{n+1}^V(j)}{\left(\boldsymbol{P}(\tau) \, \text{diag}(\boldsymbol{w}) \, \boldsymbol{h}_{n+1}^V\right)(i)}, \qquad \sum_{j=1}^m \boldsymbol{P}_n^h(i,j) = 1 \tag{12}$$

Increasing the value of $\alpha$ used to compute $\boldsymbol{V}$ strengthens the regularization, further discouraging paths through low-density states. Setting $\alpha = 0$ recovers the standard Doob kernel without density adjustment.

**Proposition 3.1** (ScooBDoob yields the target end state for one-hot $\nu$). *Assume the terminal distribution is the one-hot vector $\nu = \boldsymbol{e}_z$ concentrating all mass on a fixed target microstate $z$. Let $\boldsymbol{h}_n$ (or $\boldsymbol{h}_n^V$) be defined by the backward recursions above and let $\boldsymbol{P}_n^h$ be the corresponding Doob kernels. For any initial distribution $\mu_0$ supported on $\{i : \boldsymbol{h}_0(i) > 0\}$, the forward evolution*

$$\mu_{n+1} = \mu_n \, \boldsymbol{P}_n^h, \qquad n = 0, 1, \ldots, N-1$$

*at terminal time satisfies $\mu_N = \nu$.*

**Stiefel Manifold Constraint for Large Systems**   To enable stable eigendecomposition of the transition matrix and enforce orthonormality of the learned eigenvectors, we begin by constructing the symmetrized form of the Markov State Model. Let $\boldsymbol{P}(\tau) \in \mathbb{R}^{m \times m}$ have stationary distribution $\boldsymbol{\pi}$ with $\boldsymbol{D} = \text{diag}(\boldsymbol{\pi})$, and define the reversible symmetrization

$$\boldsymbol{M} = \boldsymbol{D}^{1/2}\boldsymbol{P}(\tau)\boldsymbol{D}^{-1/2} = \boldsymbol{Q}_{\text{MSM}}\boldsymbol{\Lambda}_{\text{MSM}}\boldsymbol{Q}_{\text{MSM}}^\top, \tag{13}$$

where $\boldsymbol{Q}_{\mathrm{MSM}} \in \mathbb{R}^{m \times r}$ has orthonormal columns and $\boldsymbol{\Lambda}_{\mathrm{MSM}} = \mathrm{diag}(\lambda_1, \ldots, \lambda_r)$ collects the top $r$ modes ($r = m$ gives the full basis). We maintain orthonormal columns by constraining $\boldsymbol{Q}_{MSM}$ to the Stiefel manifold $\mathcal{S}_{m,r}$ defined by

$$\boldsymbol{Q}_{MSM} \in \mathcal{S}_{m,r} = \{\boldsymbol{Q} \in \mathbb{R}^{m \times m} \mid \boldsymbol{Q}^\top \boldsymbol{Q} = \boldsymbol{I}_r\}, \tag{14}$$

After a Euclidean update on a chosen objective function $\mathcal{L}$,

$$\boldsymbol{Q}^{(t+1)} = \boldsymbol{Q}_{\mathbf{MSM}}{}^{(t)} - \eta \nabla_{\boldsymbol{Q}_{\mathbf{MSM}}} \mathcal{L}(\boldsymbol{Q}_{\mathbf{MSM}}{}^{(t)}), \tag{15}$$

with $\eta$ being a step size. We retract back to $\mathcal{S}_{m,r}$ via a SVD:

$$\boldsymbol{Q}^{(t+1)} = \boldsymbol{U}\boldsymbol{\Sigma}\boldsymbol{V}^\top, \quad \boldsymbol{U}^\top \boldsymbol{U} = \boldsymbol{V}^\top \boldsymbol{V} = \boldsymbol{I} \tag{16}$$

where $\boldsymbol{\Sigma} = \mathrm{diag}(\sigma_1, \ldots, \sigma_k)$. This ensures that the transition matrix $\boldsymbol{Q}_{\mathrm{MSM}}$ remains orthonormal while steering the kinetics towards the target state.

### 3.3 Learning Transition Dynamics from Markov State Models

To learn the optimal discrete bridges over microstates, we treat the empirical MSM transition matrix $\boldsymbol{P}_{\mathrm{ref}}(\tau) \in \mathbb{R}^{m \times m}$ as the fixed reference dynamics. We match the reference dynamics with a one-step parameterized network $\boldsymbol{P}_\theta(\tau)$ that is row-stochastic, and define a sequence of time-dependent tilted transition matrices with Doob's $h$-transform. Finally, we learn a time-dependent network that predicts the tilted transition probabilities $\boldsymbol{P}_{\theta,n}^h$ which preserve the MSM structure via a Stiefel manifold constraint. The full training procedure is provided in Algorithm 2.

**Parameterization of the Discrete Transition Matrices** Let $\boldsymbol{P}_\theta(\tau) \in \mathbb{R}^{m \times m}$ be the learned one-step transition matrix. Let $z_\theta$ be a neural network that produces a positive endpoint potential; define

$$\boldsymbol{h}_{\theta,N} = \nu, \quad \boldsymbol{h}_{\theta,n} = \boldsymbol{P}_\theta(\tau)\mathrm{diag}(\boldsymbol{w})\boldsymbol{h}_{\theta,n+1}, \quad n \in \{N-1, \ldots, 0\}. \tag{17}$$

with density-aware regularization $w_j = \exp(-\tau \boldsymbol{V}(j))$ introduced in Section 3.2. The student's time-inhomogeneous kernels are the discrete Doob tilts:

$$\boldsymbol{P}_{\theta,n}^h(i,j) = \frac{\boldsymbol{P}_\theta(i,j) w_j \boldsymbol{h}_{\theta,n+1}(j)}{(\boldsymbol{P}_\theta(\tau)\mathrm{diag}(\boldsymbol{w})\boldsymbol{h}_{\theta,n+1})(i)}, \qquad \sum_{j=1}^m \boldsymbol{P}_{\theta,n}^h(i,j) = 1. \tag{18}$$

### 3.4 Defining the Training Objective

**Unconditional MSM Loss** Let $C_{ij}(\tau)$ denote the empirical transition counts at lag $\tau$ and $\boldsymbol{P}_\theta(\tau) \in \mathbb{R}^{m \times m}$ be a parameterized network. We train $\boldsymbol{P}_\theta$ with an unconditional MSM loss $\mathcal{L}_{\mathrm{MSM}}$ defined as

$$\mathcal{L}_{\mathrm{MSM}} = -\sum_{i,j} C_{ij}(\tau) \log \boldsymbol{P}_{\theta,ij}(\tau) + \gamma_{\mathrm{CK}} \mathcal{L}_{\mathrm{CK}} + \gamma_{\mathrm{rev}} \mathcal{L}_{\mathrm{rev}} \tag{19}$$

$$\mathcal{L}_{\mathrm{CK}} = \sum_{k=2}^K \left\| \widehat{\boldsymbol{P}}(k\tau) - \boldsymbol{P}_\theta(\tau)^k \right\|_F^2, \quad \mathcal{L}_{\mathrm{rev}} = \left\| \boldsymbol{D}_\theta \boldsymbol{P}_\theta(\tau) - \boldsymbol{P}_\theta(\tau)^\top \boldsymbol{D}_\theta \right\|_F^2 \tag{20}$$

where $\boldsymbol{\pi}_\theta^\top \boldsymbol{P}_\theta(\tau) = \boldsymbol{\pi}_\theta^\top$ and $\boldsymbol{D}_\theta = \mathrm{diag}(\boldsymbol{\pi}_\theta)$, and $K \in \{2, 3\}$ in practice. The first term represents the count likelihood, and the second and third terms are the Chapman-Kolmogorov (CK) consistency and reversibility, respectively.

**Schrödinger Bridge Loss** To train $\boldsymbol{P}_{\theta,n}^h$ such that it predicts the optimal Schrödinger bridge defined by tilting the reference dynamics with Doob's $h$-transform, we minimize a KL-divergence-based bridge loss $\mathcal{L}_{\mathrm{bridge}}$ defined as

$$\mathcal{L}_{\mathrm{bridge}} = \sum_{n=0}^{N-1} \sum_{i=1}^m \mathrm{KL}\left(\boldsymbol{P}_{\mathrm{ref},n}^h(i,\cdot) \big\| \boldsymbol{P}_{\theta,n}^h(i,\cdot)\right) \tag{21}$$

where $\boldsymbol{P}_{\mathrm{ref},n}^h(i,\cdot)$ is defined with (12) using the fixed reference dynamics $\boldsymbol{P}_{\mathrm{ref}}(\tau)$

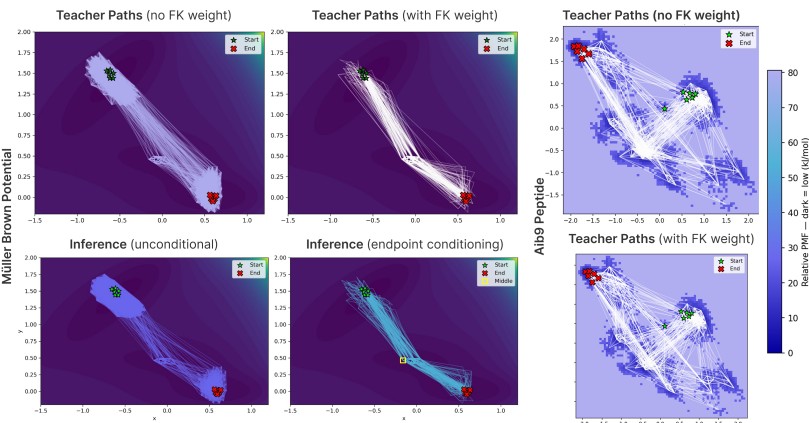

Figure 2: **Transition Paths Predicted by ScooBDoob on Müller-Brown Potential and Aib9 Peptide.** We show the teacher paths with and without density-aware FK reweighting. The inference paths are shown for the MB potential with and without endpoint conditioning.

**Stiefel Loss** Given the parameterized transition matrix $\boldsymbol{P}_\theta(\tau)$, we obtain the top-$r$ eigenvector-eigenvalue pairs by symmeterizing and diagonalizing $\boldsymbol{M}_\theta \approx \boldsymbol{Q}_\theta \boldsymbol{\Lambda}_\theta \boldsymbol{Q}_\theta^\top$, $\boldsymbol{Q}_\theta \in \mathbb{R}^{m \times r}$ (See Appendix B.1 for full details). To ensure that the eigenvectors are orthonormal, we add a soft Stiefel loss $\mathcal{L}_{\text{stief}}$ defined as

$$\mathcal{L}_{\text{stief}} = \left\| \boldsymbol{Q}_\theta^\top \boldsymbol{Q}_\theta - \boldsymbol{I}_r \right\|_F^2 + \eta \sum_{i=1}^r \max(0, |\lambda_{\theta,i}| - 1)^2 \tag{22}$$

We show in Figure A1 that the orthonormal constraint is enforced throughout training, highlighting that our approach effectively preserves the validity of the learned transition matrix. Finally, we define the **total training loss** to be the weighted sum of the MSM loss, bridge loss, and Stiefel loss, given by

$$\mathcal{L}_{\text{total}} = \mathcal{L}_{\text{MSM}} + \gamma_{\text{bridge}} \mathcal{L}_{\text{bridge}} + \gamma_{\text{stief}} \mathcal{L}_{\text{stief}} \tag{23}$$

which jointly optimizes the one-step transition matrix (teacher model) and the time-varying conditional transition dynamics.

### 3.5 SIMULATING THE LEARNED TRANSITION DYNAMICS

**Simulating Unconditional Dynamics** Given an initial distribution over microstates $\mu_0 \in \Delta^{m-1}$, we can simulate the unconditional trajectory over time $T = N\tau$ with the learned reference transition matrix $\boldsymbol{P}_\theta(\tau)$.

$$\boldsymbol{\mu}_n = \boldsymbol{\mu}_0 \boldsymbol{P}_\theta(\tau)^n, \quad \boldsymbol{x}_{n+1} \sim \boldsymbol{P}_\theta(\tau)(\boldsymbol{x}_n, \cdot) \tag{24}$$

**Target Conditioned Bridge** Given a time horizon $T = N\tau$ and a target distribution $\nu \in \mathbb{R}^m$, we can sample the intermediate trajectory from an initial distribution $\boldsymbol{\mu}_0 \in \Delta^{m-1}$

$$\boldsymbol{\mu}_{n+1} = \boldsymbol{\mu}_n \boldsymbol{P}_{\theta,n}^h \quad \text{or} \quad \boldsymbol{x}_{n+1} \sim \boldsymbol{P}_{\theta,n}^{(h,V)}(\boldsymbol{x}_n, \cdot). \tag{25}$$

over time steps $n \in \{1, \ldots, N\}$. Unconditional and target-conditioned simulation proceeds via Algorithm 3.

## 4 EXPERIMENTS

Here, we demonstrate the effectiveness of **ScooBDoob** on predicting discrete transitions between MD states conditioned on a target state. We start with a synthetic example on the Müller-Brown (MB) potential energy landscape, illustrating the model's ability to capture intermediate states between conditioned endpoints. Then, we scale our evaluation to the 9-residue $\alpha$-helical Aib9 peptide with two distinct intermediate paths (Karle and Balaram, 1990).

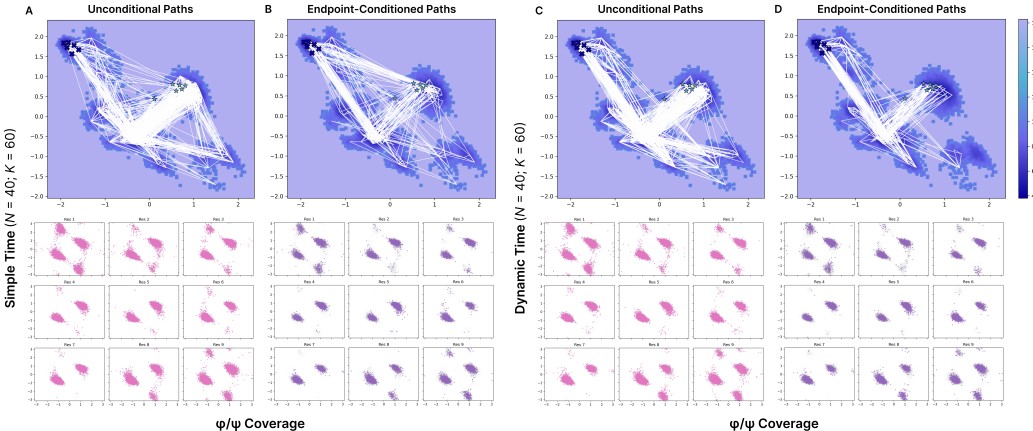

Figure 3: **Simulated transition paths for Aib9 Peptide.** Simple time indicates a fixed lag time $\tau = 60$, and dynamic time indicates a non-fixed lag time, $N$ indicates the number of jumps simulated at inference, and $K$ denotes the number of discrete microstates. Dark purple indicates high probability mass, and light purple indicates low probability mass. The axes are the $\varphi/\psi$ dihedral angles. **(A)** Unconditional paths simulated for 60 jumps. **(B)** Endpoint-conditioned paths with Doob's $h$-transform simulated with $\tau = 60$. **(C)** Unconditional paths and **(D)** endpoint-conditioned paths simulated from dynamic $\tau$.

### 4.1 MÜLLER-BROWN POTENTIAL

**Setup**  Following (Müller and Brown, 1979), we build up the testing system for a 2D Müller-Brown potential with a potential energy landscape $U(\boldsymbol{x})$ with three local minima states. We generated 8 unconditioned rollouts of length 8000 steps each, seeding half of the trajectories near the starting point at $(-0.6, 1.5)$, and the other half near the end point at $(0.6, 0.0)$. 64K frames are used for training in total. Experiment details are given in Appendix E.

**Results**  Using TICA with lag $\tau = 120$ steps, we discretize trajectories into $K = 200$ microstates. The reference MSM exhibits a spectral gap with $\lambda_2 = 0.986$, indicating slow transitions. We first constructed our FK-Doob teacher with density-aware weights and committor-based biasing to enable transition paths to cross saddle points. After training a parameterized student kernel, we found that the unconditional path explored essential states, while conditioning on the target end state significantly reduced the search space and produced more concentrated endpoints (Figure 3, Table 1). To test generalization, we evaluated zero-shot performance across $N \in \{5, 55, 120\}$. The

Table 1: **Results for Müller Brown potential with $N = 60$.**

| Metric | MB potential |
|---|---|
| Row-KL ($\downarrow$) | 0.3204 |
| CK ($\downarrow$) | 0.5401 |
| Mean $\mathcal{W}_2$ ($\downarrow$) | 0.2038 |

student kernel adapted well for $N = 55$ and $N = 120$, but for the extreme case of $N = 5$, it failed to consistently cross saddle points. This suggests that careful selection of $N$ is critical to ensure sufficient exploration time (Figure A3).

### 4.2 AIB9 PEPTIDE

**Setup**  Aib9 peptide is a 9-residue peptide experimentally validated to have two known macro intermediate states. We retrieve the Aib9 peptide trajectory with 100 ns simulation length from (Wang and Tiwary, 2021). We picked the replica at $400K$ for training and $412K$ for zero-shot prediction. There are a total of 50K frames, and 70% are used for training.

**Results**  We selected the $400K$ replica as our training trajectory and built ScooBDoob on the TICA-projected microstates of the Aib9 peptide using $\tau$ sweeping based on a balance selection from the spectral gap against CK error. The teacher kernel, trained with FK constraints, produced smoother and more connected transition paths (Figure 3). In addition to matching MSM metrics (Table 2), we mapped transitions back to the original $\psi/\phi$ angle distributions and verified that the correct intermediate states participated in the transitions with high probability.

Table 2: **Ablation studies for Aib9 experiment hyperparameters.** Metrics are computed for inference rollouts. Total steps $N$ determine the step choices for the models. The number of nearest neighbors $K$ determines the number of discrete microstates that can be transitioned into. Simple timestep defines a rigid number of steps, and dynamic timesteps allow various timesteps.

| Hyperparameter | Row-KL ($\downarrow$) | CK ($\downarrow$) | Mean $\mathcal{W}_2$ ($\downarrow$) | KL$_{\text{endpoint}}$ ($\downarrow$) |
|---|---|---|---|---|
| **Total Steps $N$ at fixed $K = 40$** | | | | |
| $N = 20$ | $0.48 \pm 0.08$ | 0.67 | 0.032 | $1.09 \pm 0.01$ |
| $N = 40$ | $0.40 \pm 0.06$ | 0.66 | 0.19 | $11.17 \pm 0.04$ |
| $N = 60$ | $0.46 \pm 0.08$ | 0.67 | 0.053 | $0.88 \pm 0.02$ |
| **Nearest Neighbor $K$ at fixed $N = 60$** | | | | |
| $K = 40$ | $0.45 \pm 0.08$ | 0.64 | 0.05 | $0.82 \pm 0.02$ |
| $K = 60$ | $0.42 \pm 0.07$ | 0.67 | 0.19 | $10.78 \pm 0.04$ |
| $K = 100$ | $0.48 \pm 0.08$ | 0.63 | 0.040 | $0.86 \pm 0.02$ |
| **Timestep $N = 40, K = 60$** | | | | |
| Simple | $0.26 \pm 0.07$ | 0.49 | 0.0008 | $0.42 \pm 0.01$ |
| Dynamic | $0.46 \pm 0.08$ | 0.67 | 0.053 | $0.88 \pm 0.02$ |

The lag time $\tau$ and the number of clusters $K$ contribute most strongly to the quality of the sampled paths. With a larger $K$, the model has more possible microstate transitions, which increases its ability to reach rare states. Conversely, a smaller $K$ makes the number of microstates closer to the number of macrostates, effectively coarsening the dynamics. With the additional of the desired end state signal during inference, the paths will guarantee to end at the endpoints, as demonstrated in the Figure 3. When comparing fixed and dynamic timesteps, the dynamic variant produces paths that visually follow the teacher more closely, often finding multiple reasonable routes to the endpoint. By contrast, the fixed variant sometimes finds the shortcut to the endpoints and involve some looping between nearby states, as shown by the higher density of the white paths chosen between some states. Although the fixed $N$ yields lower row-KL and endpoint KL numerically, these scores often reflect confident transitions rather than a more faithful path exploration follows the teacher.

In addition, we noticed that during inference, when $N = K$, the end point KL divergence spikes, even though other metrics remain comparable. We suspect that the model effectively compressed the dynamics so that the probability mass arrives at the endpoints either too quickly or along the wrong support. This creates an artificial divergence in the endpoint distribution, even though the rollout paths still appear reasonable.

To monitor spectral stability during training, we evaluated the leading eigenvalues and eigenvectors of the learned transition matrix $P_\theta$ at each epoch (Table A2, Figure A1). The dominant eigenvalue $\lambda_1$ remained near 1, as required for MSMs, and eigenvalue equation residuals were negligible, confirming numerical accuracy. Successive eigenvectors showed overlaps approaching 1 with small Frobenius distances, indicating smooth evolution and stable slow modes. These diagnostics confirm that Stiefel regularization stabilizes the eigendecomposition during training.

We also tested zero-shot generalization on replicas at $412K$ and $503K$ (Table A1). The fixed-$N$ kernel achieved lower row-KL under temperature shift by concentrating transitions into sharper steps, while the multi-$N$ kernel spread probability more smoothly across paths. This smoothing raised row-KL slightly but kept CK error low, showing that multi-$N$ training preserves overall kinetics and yields more robust rollouts at unseen temperatures despite less favorable local scores.

## 5    CONCLUSION

We have introduced **Schrö**dinger **B**ridge with **Doob**'s $h$-Transform (**ScooBDoob**), a machine learning framework for modeling molecular dynamics trajectories by learning discrete transitions between metastable states. ScooBDoob constructs a principled Schrödinger bridge from empirical MSMs using Doob's transform and density-aware regularization, enabling rare-event trajectory generation without a known energy landscape. This approach allows conditioning on endpoint structures, making it well-suited for applications like protein refolding, allosteric modulation, and conformational control. Our future extensions will incorporate experimental intermediates or kinetic priors as constraints, enabling multi-objective control over long-timescale dynamics in undersampled or sparse regimes.

## REPRODUCIBILITY STATEMENT

We have taken care to make all results in this paper reproducible. Full model descriptions, objectives, and algorithms are provided in Sections 3–3.1, with pseudocode for training and inference given in Algorithms 2 and 3. Experimental details for the Müller-Brown potential and Aib9 peptide setups are described in Sections 4, 4.2, and Appendix E. Hyperparameters, loss functions, and evaluation metrics are presented in Appendix E (Tables A3, 2, and A1). Spectral stability diagnostics are provided in Appendix D. Random seeds, training settings, and optimization hyperparameters are explicitly reported. Upon publication, we will release an repository containing all code, preprocessing scripts, model checkpoints, and one-command runners to regenerate the figures and tables. Together, these ensure full reproducibility of the results.

## ETHICS STATEMENT

This paper introduces a computational framework for simulating metastable dynamics in molecular systems. All experiments use either synthetic toy landscapes (Müller-Brown potential, Section 4) or publicly available molecular dynamics datasets (Aib9 peptide trajectory, Section 4.2, Appendix E). No human subjects, identifiable personal data, or animal studies are involved. The work is limited to methodological development and does not provide experimental protocols or actionable therapeutic targets. All datasets are cited with attribution and used under their original licenses. While generative frameworks for biomolecular dynamics could in principle have broader applications, the scope here is fundamental simulation methodology, with minimal risk of misuse. We also report compute assumptions and hyperparameters (Appendix E) to promote efficient and transparent replication.

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

## OUTLINE OF APPENDIX

In Appendix B, we discuss the construction of the Markov State Model and the Stiefel manifold constraint (B.1). Appendix E provides the setup for our experiments and the evaluation metrics. Finally, the pseudocode for training ScooBDoob is given in Appendix F.

**Notation**  In this work, we consider a molecular system with features $\boldsymbol{x}(t) \in \mathbb{R}^d$ which can be reduced with TICA to a lower-dimensional feature vector $\boldsymbol{y}(t)$. We denote the number of microstates as $m$ and the unconditional transition matrix with lag time $\tau$ as $\boldsymbol{P}(\tau) \in \mathbb{R}^{m \times m}$, where $\boldsymbol{P}_{ij}(\tau) \in \mathbb{R}$ is the probability of transition from microstate $i$ at time $t$ to microstate $j$ at time $t + \tau$. This matrix is constructed from the observed transition counts $C(i, j; \tau) \in \mathbb{R}$. The initial discrete distribution over the microstates is denoted $\mu_0 \in \Delta^{m-1}$ and the final distribution $\nu \in \Delta^{m-1}$. Given the number of lag steps $N$ with a total time horizon $T = N\tau$, we define the discrete distribution for step $n \in \{N - 1, \ldots, 0\}$ starting from $\boldsymbol{h}_N = \nu \in \Delta^{m-1}$ as $\boldsymbol{h}_n = \boldsymbol{P}(\tau)\boldsymbol{h}_{n+1}$. The Doob-tilted transition matrices given the backward distributions $\boldsymbol{h}_n$ at each time step $n$ is denoted $\boldsymbol{P}_n^h \in \mathbb{R}^{m \times m}$ where the transition probability from $i$ to $j$ is $\boldsymbol{P}_n^h(i, j) \in \mathbb{R}$. $\boldsymbol{V} : \{1, \ldots, m\} \to \mathbb{R}_{\geq 0}$ denotes a density-aware potential for each micro-state which is used to compute a weight vector $\boldsymbol{w} \in \mathbb{R}^m$ where each element is $\boldsymbol{w}(j) = \exp(-\tau \boldsymbol{V}(j))$.

The parameterized unconditional transition matrix with parameters $\theta$ is denoted $\boldsymbol{P}_\theta(\tau) \in \mathbb{R}^{m \times m}$ and the corresponding tilted distribution at step $n$ is denoted $\boldsymbol{h}_{\theta,n} \in \mathbb{R}^m$ which constructs the tilted transition matrix $\boldsymbol{P}_{\theta,n}^h \in \mathbb{R}^{m \times m}$. To define the Stiefel constraint, we symmetrize $\boldsymbol{P}_\theta(\tau)$ with the diagonal matrix $\boldsymbol{D} = \text{diag}(\boldsymbol{\pi})$ where $\boldsymbol{\pi} \in \Delta^{m-1}$ is the stationary distribution $\boldsymbol{\pi}^\top \boldsymbol{P}_\theta(\tau) = \boldsymbol{\pi}$ to get the symmetrical matrix $\boldsymbol{M} \in \mathbb{R}^{m \times m}$. Then, we perform a symmetric eigendecomposition to obtain the orthonormal matrix $\boldsymbol{Q}_{\text{MSM}} \in \mathbb{R}^{m \times r}$ and eigenvalues $\boldsymbol{\Lambda}_{\text{MSM}} = \text{diag}(\lambda_1, \ldots, \lambda_r)$. At inference time, we generate intermediate distributions $\mu_n \in \Delta^{m-1}$ by applying the learned transition to the initial distribution $\mu_0$ and sampling discrete states $\boldsymbol{x}_n \sim \mu_n$.

## A  RELATED WORKS

**Transition Path Sampling (TPS)**  Computational approaches to transition path sampling over energy landscapes have been widely explored (Bolhuis et al., 2002; Dellago et al., 1998; Vanden-Eijnden et al., 2010). Traditionally, non-ML approaches have leveraged low-dimensional representations of molecules via *collective variables* (CVs) (Hooft et al., 2021), including steered MD (Schlitter et al., 1994; Izrailev et al., 1999), umbrella sampling (Torrie and Valleau, 1977; Kästner, 2011), meta-dynamics (Laio and Parrinello, 2002; Ensing et al., 2006; Branduardi et al., 2012; Bussi and Branduardi, 2015), adaptive biasing force Comer et al. (2015), and on-the-fly probability-enhanced sampling (Invernizzi and Parrinello, 2020). Such methods are powerful when good CVs are known, but selecting CVs remains challenging (Hooft et al., 2021).

**State-based Kinetic Models**  An alternative line of work focuses on *state-based* models that extract slow kinetics directly from simulation data. A rigorous theory shows that optimal CVs correspond to the eigenfunctions of the transfer operator underlying MD (Noé and Clementi, 2017). Practical approximations include Time-lagged Independent Component Analysis (TICA) (Pérez-Hernández et al., 2013), Diffusion Maps (Coifman et al., 2005), and Markov State Models (MSMs) (Prinz et al., 2011; Bowman et al., 2014; Mardt et al., 2018), which discretize conformational space into metastable states and estimate transition probabilities. These approaches unify dimensionality reduction and kinetics estimation under a variational principle.

**Modeling Molecular Dynamics**  More recently, coarse-grained and full-atom generative models have sought to reconstruct trajectories and sample new transitions (Arts et al., 2023; Charron et al., 2025; Kohler et al., 2023; Majewski et al., 2023; Lu et al., 2025; Raja et al., 2025). Methods such as score-based modeling (Daigavane et al.; Tan et al., 2025), energy-based modeling (Lu et al., 2025; Lewis et al., 2025b), and flow-based generative dynamics (Jing et al., 2024; Kohler et al., 2023; Rehman et al., 2025) attempt to bypass explicit force fields by directly learning mappings between metastable ensembles.

**Schrödinger Bridge over MSM** In discrete time, classical Schrödinger Bridges (SBs) on Markov chains (Beghi, 2002; Pavon et al., 2010; Pavon and Ticozzi, 2010) solve endpoint-constrained path-space maximum-entropy problems by a multiplicative Doob $h$-transform of a prior kernel, with potentials given by space-time harmonic functions and uniqueness via the Beurling-Jamison theorem. While this theory provides constructive formulas for tilting Markov kernels analytically, **ScooBDoob** learns these tiltings directly from molecular dynamics trajectories. The student kernel $\boldsymbol{P}_\theta$ acts as a parametric Doob $h$-transform, constrained by reversibility and conditioned on metastable endpoints.

Whereas SBs enforce exact marginals and admit closed-form harmonic potentials, **ScooBDoob** enforces macrostate marginals and learns an approximate bridge distribution, extending the maximum entropy principle into a data-driven regime. Thus, given an MSM prior $\prod$ and endpoint marginals $(\mu_0, \mu_N)$, with $\mu_N$ concentrated on a target macrostate, ScooBDoob seeks a Markov bridge kernel $\boldsymbol{P}_\theta$ that approximates the SB minimizer $\mathbb{P}^{\mathrm{SB}} = \min_{\mathbb{P}}\{\mathrm{KL}(\mathbb{P}\|\mathbb{Q}) : \mathbb{P}_0 = \mu_0, \mathbb{P}_N = \mu_N\}$ by training $\boldsymbol{P}_\theta$ to match the Doob-tilted optimum through MSM consistent losses.

**Learning Schrödinger Bridges** Schrödinger bridge methods have also been used outside molecular dynamics to construct samplers or generative models in continuous time. Bernton et al. (2019) approximates iterative proportional fitting in continuous state spaces to reduce variance in Annealed Importance Sampling and Sequential Monte Carlo. With others (De Bortoli et al., 2021; Liu et al., 2023) connect bridge dynamics with score-based generative modeling and Kim et al. (2024) extend the scope into graph transformation. While these methods focus on sampling from static or structured distributions via continuous or discrete diffusions, our approach differs in that we operate on finite-state MSMs and learn bridge kernels directly from MD trajectories to generate endpoint-conditioned paths between metastable states.

## B EXTENDED THEORETICAL BACKGROUND

Here, we describe preliminaries and additional details on the theory of Schrödinger bridge matching with Doob's $h$-Transform and optimization on the Stiefel manifold.

### B.1 STIEFEL MANIFOLD CONSTRAINT

**Stiefel Manifold** The Stiefel manifold, denoted $\mathcal{S}_{n,k}$ is the set of $n \times k$ ($n \geq k$) orthonormal rectangular matrices defined as

$$\mathcal{S}_{n,k} = \{\boldsymbol{Q} \in \mathbb{R}^{n \times k} | \boldsymbol{Q}^\top \boldsymbol{Q} = \boldsymbol{I}_k\} \tag{26}$$

where $\boldsymbol{I}_k \in \mathbb{R}^{k \times k}$ is the $k \times k$ identity matrix.

**Eigendecomposition of MSM** Given the MSM transition matrix $\boldsymbol{P}(\tau)$ at lag time $\tau$, there exists a *stationary distribution* $\boldsymbol{\pi} \in \mathbb{R}^m$ such that $\boldsymbol{P}(\tau)\boldsymbol{\pi} = \boldsymbol{\pi}$ (Beauchamp et al., 2011). Symmetrize via:

$$\boldsymbol{M} = \boldsymbol{D}^{\frac{1}{2}} \boldsymbol{P}(\tau) \boldsymbol{D}^{-\frac{1}{2}}, \quad \boldsymbol{D} = \mathrm{diag}(\boldsymbol{\pi}) \tag{27}$$

We then diagonalize

$$\boldsymbol{M} = \boldsymbol{Q}_{\mathrm{MSM}} \boldsymbol{\Lambda}_{\mathrm{MSM}} \boldsymbol{Q}_{\mathrm{MSM}}^\top, \quad \boldsymbol{Q}_{\mathrm{MSM}} \in \mathbb{R}^{m \times r}, \ \boldsymbol{\Lambda}_{\mathrm{MSM}} = \mathrm{diag}(\lambda_1, \ldots, \lambda_m), \tag{28}$$

so that

$$\boldsymbol{P}(\tau) = \boldsymbol{D}^{-\frac{1}{2}} \boldsymbol{Q}_{\mathrm{MSM}} \boldsymbol{\Lambda}_{\mathrm{MSM}} \boldsymbol{Q}_{\mathrm{MSM}}^\top \boldsymbol{D}^{\frac{1}{2}} = \sum_{i=1}^m \lambda_i \, r_i \, \ell_i^\top, \tag{29}$$

where $\lambda_i$ are the eigenvalues of $\boldsymbol{P}(\tau)$, and $r_i$ and $\ell_i$ are the corresponding right and left eigenvectors, respectively. $r_i = \boldsymbol{D}^{-\frac{1}{2}} \boldsymbol{q}_i$, $\ell_i = \boldsymbol{D}^{\frac{1}{2}} \boldsymbol{q}_i$, and bi-orthogonality $\ell_i^\top \boldsymbol{D} r_j = \delta_{ij}$ maintains.

### B.2 CHAPMAN-KOLMOGOROV CONSISTENCY

For a Markov chain modeling of dynamics at lag time $\tau$, one step of length $k\tau$ should look the same as $k$ consecutive steps of length $\tau$.

$$\boldsymbol{P}(k\tau) = \boldsymbol{P}(\tau)^k, \quad k = 2, 3, \ldots \tag{30}$$

If such CK consistency fails, the assumption that the dynamics are approximately Markovian at lag $\tau$ does not hold.

From the MD trajectory, count matrix $C_{ij}$ can be built to represent jump from state $i$ to state $j$ after lag $\tau$. Assume that at lag $k\tau$ there exists $C_{k\tau}$, counts can be turned into probability:

$$\hat{P}_{\text{ref}}(\tau)[i,j] = \frac{C_\tau[i,j]}{\sum_{j'}(C_\tau[i,j'])} \tag{31}$$

where $j = k$ depend on the metrics asked. ScoobDoob parameterizes the one-step kernel $P_\theta$, and the CK consistency is used to measure whether

$$\boldsymbol{P}_\theta^k \approx \hat{\boldsymbol{P}}_{\text{ref}}(k\tau). \tag{32}$$

Low CK error indicates that the learned one-step dynamics compose correctly over longer lags, a prerequisite for stable implied timescales and reliable kinetic predictions (Prinz et al., 2011; Bowman et al., 2014; Noé and Clementi, 2017).

## C  THEORETICAL PROOFS

**Lemma 1** (Row-stochasticity and telescoping identity). *Let $\boldsymbol{P}(\tau) \in \mathbb{R}^{m \times m}$ be a row-stochastic MSM transition matrix and let $(\boldsymbol{h}_n)_{n=0}^N$ be the backward sequence defined by*

$$\boldsymbol{h}_N = \nu, \qquad \boldsymbol{h}_n = \boldsymbol{P}(\tau)\,\boldsymbol{h}_{n+1}, \quad n = N-1, \ldots, 0,$$

*or, in the density-aware case, by*

$$\boldsymbol{h}_N^V = \nu, \qquad \boldsymbol{h}_n^V = \boldsymbol{P}(\tau)\,\text{diag}(\boldsymbol{w})\,\boldsymbol{h}_{n+1}^V, \quad n = N-1, \ldots, 0,$$

*with $w_j = \exp(-\tau V(j))$ as in Section 3.2. Define the time-inhomogeneous Doob kernels*

$$\boldsymbol{P}_n^h(i,j) = \frac{\boldsymbol{P}_{ij}(\tau)\,\boldsymbol{h}_{n+1}(j)}{\boldsymbol{h}_n(i)}, \quad \text{or} \quad \boldsymbol{P}_n^h(i,j) = \frac{\boldsymbol{P}_{ij}(\tau)\,w_j\,\boldsymbol{h}_{n+1}^V(j)}{\big(\boldsymbol{P}(\tau)\,\text{diag}(\boldsymbol{w})\,\boldsymbol{h}_{n+1}^V\big)(i)}.$$

*Then for every $n$ and $i$, $\sum_j \boldsymbol{P}_n^h(i,j) = 1$ (row-stochasticity). Moreover, for any path $(i_0, \ldots, i_N)$ the path probability under the Doob chain satisfies the telescoping identity*

$$\mu_0(i_0) \prod_{n=0}^{N-1} \boldsymbol{P}_n^h(i_n, i_{n+1}) = \mu_0(i_0) \left( \prod_{n=0}^{N-1} \boldsymbol{P}_{i_n i_{n+1}}(\tau) \right) \cdot \frac{\boldsymbol{h}_N(i_N)}{\boldsymbol{h}_0(i_0)},$$

*with $\boldsymbol{h}_n$ replaced by $\boldsymbol{h}_n^V$ in the density-aware case. Row-stochasticity is immediate from the weighted space-time harmonic relation $h_n = P(\tau)\text{diag}(w)h_{n+1}$ (cf. Pavon-Ticozzi, Eq 25) (Pavon and Ticozzi, 2010), exactly as in their Eq. 27, where $\sum_j \hat{p}_{ij} = 1$ follows by dividing $\sum_j \pi_{ij}\varphi(t+1,j)$ by $\varphi(t,i)$.*

*Proof.* Row-stochasticity follows from the backward recursion, where we sum over columns $j$:

$$\sum_j^m \boldsymbol{P}_n^h(i,j) = \frac{1}{\boldsymbol{h}_n(i)} \sum_j^m \boldsymbol{P}_{ij}(\tau)\,\boldsymbol{h}_{n+1}(j) = \frac{\big(\boldsymbol{P}(\tau)\,\boldsymbol{h}_{n+1}\big)(i)}{\boldsymbol{h}_n(i)} = \frac{\boldsymbol{h}_n(i)}{\boldsymbol{h}_n(i)} = 1,$$

and similarly in the density-aware case with $\text{diag}(\boldsymbol{w})\,\boldsymbol{h}_{n+1}^V$. For the telescoping identity, expand the product of Doob factors and note that the ratios $\boldsymbol{h}_{n+1}(i_{n+1})/\boldsymbol{h}_n(i_n)$ cancel along the path, leaving only $\boldsymbol{h}_N(i_N)/\boldsymbol{h}_0(i_0)$. The argument is identical with $\boldsymbol{h}^V$. □

**Proposition 3.1** (ScooBDoob yields the target end state for one-hot $\nu$). *Assume the terminal distribution is the one-hot vector $\nu = \boldsymbol{e}_z$ concentrating all mass on a fixed target microstate $z$. Let $\boldsymbol{h}_n$ (or $\boldsymbol{h}_n^V$) be defined by the backward recursions above and let $\boldsymbol{P}_n^h$ be the corresponding Doob kernels. For any initial distribution $\mu_0$ supported on $\{i : \boldsymbol{h}_0(i) > 0\}$, the forward evolution*

$$\mu_{n+1} = \mu_n\,\boldsymbol{P}_n^h, \qquad n = 0, 1, \ldots, N-1$$

*at terminal time satisfies $\mu_N = \nu$.*

*Proof.* We prove the density-aware case (the unweighted case is identical with $w_j \equiv 1$). Since $\nu = e_z$, we have

$$\boldsymbol{h}_N^V = \nu = \boldsymbol{e}_z, \qquad \boldsymbol{h}_{N-1}^V = \boldsymbol{P}(\tau)\operatorname{diag}(\boldsymbol{w})\,\boldsymbol{e}_z = w_z\,\boldsymbol{P}(\tau)\,\boldsymbol{e}_z,$$

so that $\boldsymbol{h}_{N-1}^V(i) = w_z\,\boldsymbol{P}_{iz}(\tau)$ for every $i$. By Eq. (12), for the final step $n = N-1$ and any $i, j$,

$$\boldsymbol{P}_{N-1}^h(i,j) = \frac{\boldsymbol{P}_{ij}(\tau)\,w_j\,\nu(j)}{w_z\,\boldsymbol{P}_{iz}(\tau)} = \delta_{jz} = \begin{cases} 1, & j = z, \\ 0, & j \neq z. \end{cases}$$

Thus the last-step kernel $\boldsymbol{P}_{N-1}^h$ deterministically sends all mass into $z$, so regardless of $\mu_{N-1}$,

$$\mu_N(j) = \sum_i^m \mu_{N-1}(i)\,\boldsymbol{P}_{N-1}^h(i,j) = \sum_i^m \mu_{N-1}(i)\delta_{jz} = \delta_{jz} = \boldsymbol{e}_z = \nu(j).$$

where $\delta$ is the Kronecker delta. Because each $\boldsymbol{P}_n^h$ is row-stochastic (Lemma 1), normalization and positivity are preserved throughout, and the recursion is well defined for all $n$. Hence $\mu_N = \nu$. $\qquad\square$

**Remark 1** (General terminal distributions). *For a general terminal law $\nu \in \Delta^{m-1}$ (not necessarily one-hot), the Doob kernels (Eqs. (2) or (5)) still define a valid inhomogeneous Markov chain. Writing $\alpha_n(i) := \mu_n(i)/\boldsymbol{h}_n(i)$ (or $\alpha_n(i) := \mu_n(i)/\boldsymbol{h}_n^V(i)$ in the density-aware case). Then one verifies that $\alpha_{n+1}^\top = \alpha_n^\top \boldsymbol{P}(\tau)$ (unweighted), $\alpha_{n+1}^\top = \alpha_n^\top \boldsymbol{P}(\tau)\operatorname{diag}(w)$ (density-aware). Hence*

$$\mu_N(j) = \nu(j)\,[\alpha_0^\top \boldsymbol{P}(\tau)^N]_j \qquad \text{or} \qquad \mu_N(j) = \nu(j)\,[\alpha_0^\top (\boldsymbol{P}(\tau)\operatorname{diag}(w))^N]_j$$

*To enforce $\mu_N = \nu$ componentwise for arbitrary $\mu_0$, one requires the full Schrödinger system (maximum-entropy) compatibility between the boundary marginals, equivalently choosing the forward potential so that the terminal factor equals $\mathbf{1}$; see Appendix A.3 and Eq. (24). In our experiments we restrict to the one-hot terminal law (Appendix D.1), for which Proposition 3.1 applies directly.*

## D ADDITIONAL RESULTS AND DISCUSSION

Table A1: **Zero-shot evaluation across temperatures.** Metrics computed at replicas 412 K and 503 K using models trained at 400 K. Lower is better. Multi-$N$ maintains CK consistency with only a modest increase in KL.

| | 412 K | | | 503 K | | |
|---|---|---|---|---|---|---|
| Model | Row-KL | CK | $\mathcal{W}_2$ | Row-KL | CK | $\mathcal{W}_2$ |
| Fixed-$N$ | 1.462 | 0.925 | 0.016 | 1.047 | 0.869 | 0.041 |
| Multi-$N$ | 1.520 | 0.927 | 0.020 | 1.124 | 0.877 | 0.023 |

### D.1 EVALUATING SPECTRAL STABILITY DURING TRAINING

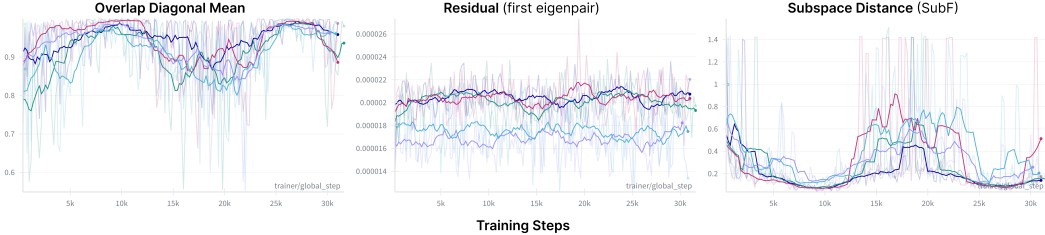

Figure A1: **Spectral stability metrics over training steps.**

**Spectral gap** Denote the symmetrized operator by $\boldsymbol{M} \in \mathbb{R}^{m \times m}$ with eigenvalues $1 = \lambda_1 \geq \lambda_2 \geq \ldots \lambda_m$. The spectral gap is the difference between $\lambda_1 - \lambda_2$, which quantifies separation between the stationary mode and the slowest dynamic process (Prinz et al., 2011). A larger gap is a clearer metastable separation.

Table A2: Spectral stability diagnostics of the learned MSM during training. All metrics indicate stable eigenvalues and eigenvectors across training, attributed to the Stiefel constraints.

| Metric | Value (mean $\pm$ std) |
|---|---|
| Spectral gap ($\mathrm{gap}_1$) | $0.091_{\pm 0.014}$ |
| Perron eigenvalue ($\lambda_1$) | $1.000028_{\pm 0.000001}$ |
| Overlap diag. mean ($\uparrow$) | $0.951_{\pm 0.084}$ |
| Overlap diag. min ($\uparrow$) | $0.829_{\pm 0.340}$ |
| Residual (first eigenpair) ($\downarrow$) | $(1.9_{\pm 0.3}) \times 10^{-5}$ |
| Residual (mean top-$r$) ($\downarrow$) | $(2.3_{\pm 0.1}) \times 10^{-5}$ |
| Subspace distance (SubF) ($\downarrow$) | $0.341_{\pm 0.494}$ |

**Perron eigenvalue**  For any row-stochastic transition matrix, the Perron-Frobenius theorem guarantees a leading eigenvalue $\lambda_1 = 1$. Deviations indicate difficulty in normalization and reversibility of $\boldsymbol{P}_\theta(\tau)$ (Smyth, 2002).

**Overlap diag.**  Let $\boldsymbol{Q}_{\mathrm{MSM}}^{(t)}$ and $\boldsymbol{Q}_{\mathrm{MSM}}^{(t-1)}$ denote the top $r$ eigenvectors of $M$ at successive training steps. The overlap matrix $\boldsymbol{O} = (\boldsymbol{Q}_{\mathrm{MSM}}^{(t-1)})^\top \boldsymbol{Q}_{\mathrm{MSM}}^{(t)}$ measures alignment (Husic and Pande, 2018). The mean and minimum of the diagonal entries of $|\boldsymbol{O}|$ indicates how stable each eigenvector is across epochs.

**Residual**  For each eigenpair $(\lambda_i, \boldsymbol{q}_i)$ with $q_i$ a column of $\boldsymbol{Q}_{\mathrm{MSM}}$, the residual is defined follows Simoncini (2005) as

$$\|\boldsymbol{M}\boldsymbol{q}_i - \lambda_i\boldsymbol{q}_i\|_2. \tag{33}$$

Small numbers confirms that the computed eigenpairs solve the eigenvalue problem accurately.

**Subspace distance**  The subspace spanned between epochs by the top $r$ eigenvectors is represented by the projection matrix $Q_{\mathrm{MSM}}Q_{\mathrm{MSM}}^\top$. Subspace distance between two consecutive steps is measured as the Frobenius norm

$$\left\|\boldsymbol{Q}_{\mathrm{MSM}}^{(t)}(\boldsymbol{Q}_{\mathrm{MSM}}^{(t)})^\top - \boldsymbol{Q}_{\mathrm{MSM}}^{(t-1)}(\boldsymbol{Q}_{\mathrm{MSM}}^{(t-1)})^\top\right\|_F, \tag{34}$$

which is invariant to rotations and sign flips. Smaller values indicate that the slow kinetic subspace is stable across training iterations.

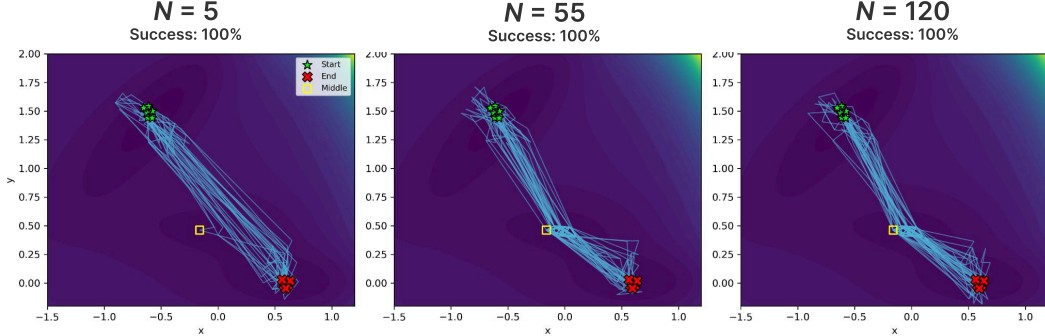

Figure A2: **Transition Paths Predicted by ScooBDoob on Müller-Brown Potential.** Tested path generation on unseen number of steps $N$. Green stars indicate starting states and red Xs indicate target end state. Intermediate transition states are marked with the yellow square.

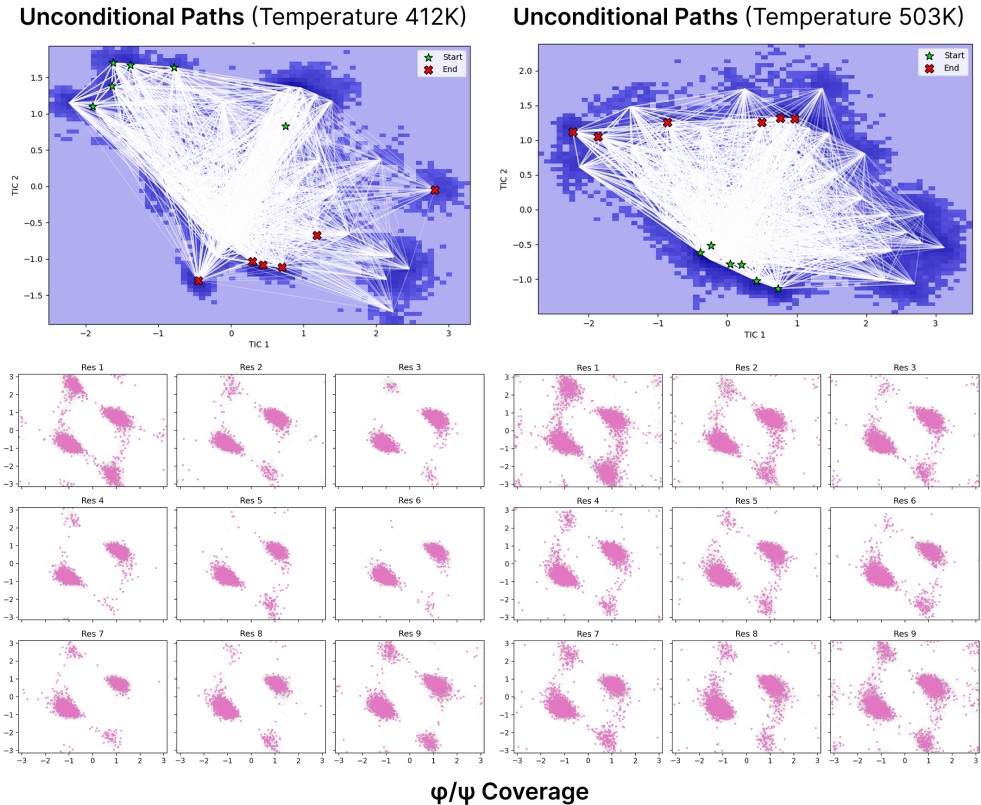

**φ/ψ Coverage**

Figure A3: **Unconditional transition paths of Aib9 peptide at temperatures of** $412K$ **and** $503K$. Simulations were performed with trained models under the identical conditions of $K = 40$ and $N = 60$. **Above**: Darker color indicates lower-energy states, with white lines showing sampled transition paths. Start and End states were determined following Appendix E. **Below**: Color highlights the states visited by the model. The grey background indicates ground-truth coverage.

# E    EXPERIMENTAL DETAILS

## E.1    AUTOMATIC $\tau$ SWEEPING AND ENDPOINT DETERMINATION

For systems like AIB9 where inputs are torsion features without clear labels for start and end microstates, we avoid manual choices and infer 1) a suitable lag time $\tau$ and 2) representative start/end state sets directly from kinetics estimated on the data.

Based on our previous discussion on MSMs, the pair $(\lambda_2, q_2)$ encodes the slowest nontrivial relaxation. We use the sign of the second eigenvector $q_2$ to produce a coarse two-well split (Röblitz and Weber, 2013):

$$\mathcal{A} = \{i : (q_2)_i \leq 0\}, \quad \mathcal{B} = \{i : (q_2)_i > 0\}. \tag{35}$$

To get confident endpoints for conditioning, we then pick the $k$ most negative entries of $q_2$ as the start set $S_{\text{start}}$ and the $k$ most positive as the end set $S_{\text{end}}$. In the current experiment, $k$ is set to 6.

For a chosen lag $\tau$, the implied timescale of the slowest process is

$$t_2(\tau) = \frac{-\tau}{\ln |\lambda_2|}, \quad \text{spectral\_gap} = 1 - |\lambda_2|. \tag{36}$$

A larger gap implied clearer separation between the stationary mode and the slowest transition, which tends to stabilize metastable assignments.

A grid of lag steps $\tau_{\text{multiple}}$ range from $40$ to $200$ was tested. The final $\tau$ will be picked by maximizing the score below that favors both kinetic separation and a split with balanced start and end states:

$$\text{score}(\tau) = \text{spectral\_gap}(\tau)(0.5 \, + \, 0.5 \cdot \frac{\min(|\mathcal{A}|, |\mathcal{B}|)}{\max(1, \max(|\mathcal{A}|, |\mathcal{B}|))}) \tag{37}$$

Then during training, at $\tau^*$ we set $S_{\text{start}}/S_{\text{end}}$ to the $k$ most negative/positive entries of $q_2$.

### E.2  Constructing the Teacher Transition Matrix

The teacher transition matrix $\boldsymbol{P}_{\text{ref}}(\tau)$ is used to define the matching objective of the parameterized student model $\boldsymbol{P}_\theta(\tau)$. We fix a time horizon $T = N\tau$ and a terminal distribution $\nu \in \Delta^{m-1}$, where $N$ is the number of lag steps. For a target state $\boldsymbol{z}$, we set the terminal distribution to the one-hot vector. We define the conditional distributions at each time step $\boldsymbol{h}_n^V \in \Delta^{m-1}$ as

$$\boldsymbol{h}_N = \nu, \qquad \boldsymbol{h}_n \, = \, \boldsymbol{P}_{\text{ref}}(\tau)\,\text{diag}(\boldsymbol{w})\,\boldsymbol{h}_{n+1}, \quad n \in \{N-1, \dots, 0\}. \tag{38}$$

The Doob $h$-transformed teacher transition matrices at each time step are then defined as

$$\boldsymbol{P}_{\text{ref},n}^h(i,j) = \frac{\boldsymbol{P}_{\text{ref}}(i,j;\tau)\boldsymbol{w}(j)\,\boldsymbol{h}_{\theta,n+1}(j)}{\left(\boldsymbol{P}_{\text{ref}}(\tau)\,\text{diag}(\boldsymbol{w})\,\boldsymbol{h}_{\theta,n+1}\right)(i)} \tag{39}$$

### E.3  Data Curation

**Synthetic Müller-Brown Potential**    Following (Müller and Brown, 1979), we build up the testing system for a 2D and 3D Müller-Brown potential with a potential energy landscape $U(\boldsymbol{x})$ with three local minima states.

$$U(\boldsymbol{x}) = \sum_{j=1}^4 A_j \cdot \exp[a_j(x_1 - X_j)^2 + b_j(x_1 - Y_j)(x_2 - X_j) + c_j(x_2 - Y_j)^2] \tag{40}$$

where $a = (-1, -1, -6.5, -0.7)$, $b = (0, 0, 11, 0.6)$, $c = (-10, -10, -6.5, 0.7)$, $A = (-200, -100, -170, 15)$, $X = (1, 0, -0.5, -1)$, $Y = (0, 0.5, 1.5, 1)$, as formulated in (Müller and Brown, 1979; Hernández et al., 2018). The dynamics are governed by

$$\dot{\boldsymbol{x}}(t) = -\beta \nabla U(\boldsymbol{x}) + \sqrt{2D}\eta(t) \tag{41}$$

where $\beta = 1$, $\eta(t)$ is Gussian noise with zero mean, time step $\Delta t = 10^{-3}$, and reflecting bounds $(-1.5, 1.2) \times (-0.2, 2.0)$.

### E.4  Loss Construction

We used a complex loss system to maintain the Markov State Model properties. Additional constraints like the Chapman-Kolmogorov loss ensure the conservation of the stationary distribution at the second-highest eigenvalue. Reversibility loss ensures that the detailed balance is held in the MSM system, and the Stiefel constraint ensures that the eigenvectors remain orthogonal.

For protein systems, the $K$ can be large with thousands of states, so we approximate the $\boldsymbol{P}_\theta$ by only predicting the transition probabilities for the $k = 48$ nearest neighbors:

$$\forall j \in \mathcal{N}(i), \;\; \boldsymbol{P}_\theta(i,j) = \text{softmax}(p_\theta(\boldsymbol{z}_i, \boldsymbol{z}_j; n)) \tag{42}$$

where $\mathcal{N}(i)$ are the k nearest neighbors of state $i$.

### E.5  Training Details

For MB potential training, a two-layer MLP with a hidden dimension of 128 and a dropout rate of 0.1 was used to map the features to a scalar score. We trained the model for 200 epochs with early stopping. The learning rate was set to $3e^{-3}$ using the Adam optimizer. All default hyperparameters are given in Table A3.

For Aib9 peptide experiments, we concatenated all angle features into a $36D$-shaped input for the model, and applied a 2-layer MLP encoder that takes paired interaction features as input. The learning rate was set to $1e^{-3}$ using the Adam optimizer. Training occurred for 200 epochs with early stopping. All default hyperparameters are given in Table A3.

Table A3: **Default hyperparameters for MB and Aib9 peptide experiments.**

| Experiment | $\Delta t$ | Temp (K) | LR | Epochs | $\lambda_{\text{CK}}$ | $\lambda_{\text{rev}}$ | $\lambda_{\text{stf}}$ | $\lambda_{\text{br}}$ | Paths (unc/cond) |
|---|---|---|---|---|---|---|---|---|---|
| MB Potential | $1.0 \times 10^{-3}$ | – | $1 \times 10^{-3}$ | 200 | 1.0 | 1.0 | 1.0 | 1.0 | 200 / 200 |
| Aib9 Peptide | 2.0 ps | 400 | $1 \times 10^{-3}$ | 200 | 1.0 | 2.0 | 2.0 | 1.0 | 100 / 100 |

### E.6   EVALUATION METRICS

**Row-KL Divergence**   We evaluate the KL-divergence of the predicted transition probabilities from each micro-state $i \in \{1, \ldots, m\}$ defined as a row of the parameterized transition matrix $\boldsymbol{P}_\theta(\tau)$ compared to the teacher transition matrix $\boldsymbol{P}_{\text{ref}}(\tau)$.

$$D_{\text{KL}}(\boldsymbol{P}_\theta(i, \cdot) || \boldsymbol{P}_{\text{ref}}(i, \cdot)) = \sum_{j=1}^{m} \boldsymbol{P}_\theta(i, j) \log \frac{\boldsymbol{P}_\theta(i, j)}{\boldsymbol{P}_{\text{ref}}(i, j)} \tag{43}$$

**Wasserstein-2 Distance ($\mathcal{W}_2$)**   We compute the $\mathcal{W}_2$ distance of the predicted terminal state

$$\mathcal{W}_2 = \left( \min_{\pi \in \Pi(p,q)} \int \|\boldsymbol{x} - \boldsymbol{y}\|_2^2 d\pi(\boldsymbol{x}, \boldsymbol{y}) \right)^{1/2} \tag{44}$$

**Chapman-Kolmogorov Error**   We are comparing the student kernel $\boldsymbol{P}_\theta$ at lag $\tau$ against empirical $2-$lag kernel from counts to satisfy:

$$\boldsymbol{P}(2\tau) = \boldsymbol{P}(\tau)^2 \tag{45}$$

The error is reported to be:

$$\frac{\left\| \boldsymbol{P}_\theta^2 - \boldsymbol{P}_{\text{ref}}(2\tau) \right\|_F}{\max(10^{-16}, \|\boldsymbol{P}_{\text{ref}}(2\tau)\|_F)} \tag{46}$$

If the error is small, the model composes correctly and respect Markovianity, otherwise the model is not consistent across time lags.

## F ALGORITHMS

Here, we provide the pseudocode for the construction of the teacher transition matrices and training the parameterized time-dependent generators in Algorithm 2 and the procedure for simulating the unconditional and target-conditioned dynamics with ScooBDoob in Algorithm 3.

---

**Algorithm 2 Training ScooBDoob**

1: **Input:** Observed count of transitions between states $i \to j$ at $\tau$ lag $C(i, j; \tau)$ for all $i, j \in \{1, \ldots, m\}$

2:

3: **while** Training **do**

4:      $\boldsymbol{P}_{ij}(\tau) \leftarrow \frac{C(i,j;\tau)}{\sum_{j'} C(i,j';\tau)}$        $\triangleright$ *compute transition probabilities from each microstate*

5:      $\boldsymbol{P}(\tau) \leftarrow [\boldsymbol{P}_{ij}(\tau)]$        $\triangleright$ *construct unconditional transition matrix*

6:      $\boldsymbol{V}(i) \leftarrow \alpha/(C_i + 1), \boldsymbol{w}(i) \leftarrow \exp(-\tau \boldsymbol{V}(i))$        $\triangleright$ *density-aware weights*

7:      $\boldsymbol{h}_N^V \leftarrow \nu$        $\triangleright$ *initialize terminal condition*

8:      **for** $n$ in $N - 1, \ldots, 0$ **do**

9:          $\boldsymbol{h}_n^V \leftarrow \boldsymbol{P}(\tau)(\text{diag}(\boldsymbol{w})\boldsymbol{h}_{n+1}^V)$        $\triangleright$ *compute tilted distributions*

10:          $\boldsymbol{P}_n^V(i, j) \leftarrow \frac{\boldsymbol{P}_{ij}(\tau)\boldsymbol{w}(j)\boldsymbol{h}_{n+1}^V(j)}{(\boldsymbol{P}(\tau)\text{diag}(\boldsymbol{w})\boldsymbol{h}_{n+1}^V)(i)}$        $\triangleright$ *compute doob-tilted probabilities*

11:          $\boldsymbol{P}_n^V \leftarrow [\boldsymbol{P}_n^V(i, j)]$        $\triangleright$ *construct matrix*

12:

13:      **end for**

14:      **for** micro-state $i$ in $1, \ldots, m$ **do**        $\triangleright$ *train generator for each state $i$*

15:          $\boldsymbol{P}_{n,\theta}^h(i, \cdot) \leftarrow \text{NN}(\theta)$

16:          Compute loss $\mathcal{L}_{\text{total}}(\theta) = \mathcal{L}_{\text{MSM}}(\theta) + \gamma_{\text{bridge}}\mathcal{L}_{\text{bridge}}(\theta) + \gamma_{\text{stief}}\mathcal{L}_{\text{stief}}(\theta)$

17:          Optimize $\theta$ with $\nabla_\theta \mathcal{L}_{\text{total}}$

18:      **end for**

19: **end while**

20: **return** parameterized transition predictor $\boldsymbol{P}_\theta(\tau) : [0, 1] \to \mathbb{R}^{m \times m}$

---

**Algorithm 3 Inference with ScooBDoob**

1: **Input:** parameterized model $\boldsymbol{P}_\theta(\tau) : [0, 1] \to \mathbb{R}^{m \times m}$, initial distribution $\mu_0$, number of steps $N$

2: $\mathcal{P} \leftarrow \{\}$        $\triangleright$ *initialize path*

3: **for** step $n$ in $1, \ldots, N - 1$ **do**

4:      $\boldsymbol{\mu}_n \leftarrow \boldsymbol{\mu}_{n-1}\boldsymbol{P}_\theta(\tau)$        $\triangleright$ *predict distribution over microstates*

5:      $\boldsymbol{z}_n \sim \boldsymbol{\mu}_n$        $\triangleright$ *sample discrete state from distribution*

6:      $\mathcal{P} \leftarrow \mathcal{P} \cup \{\boldsymbol{\mu}_n, \boldsymbol{z}_n\}$        $\triangleright$ *append to path*

7: **end for**

8: **return** $\mathcal{P}, \boldsymbol{\mu}_N, \boldsymbol{z}_n$

---

# G    USE OF LARGE LANGUAGE MODELS (LLMS)

We acknowledge the use of large language models (LLMs) to assist in polishing and editing parts of this manuscript. LLMs were used to refine phrasing, improve clarity, and ensure consistency of style across sections. All technical content, experiments, analyses, and conclusions were developed by the authors, with LLM support limited to language refinement and editorial improvements.

