# OpenReview forum: "ScooBDoob: Schrödinger Bridge with Doob’s h-Transform for Molecular Dynamics"
_ICLR.cc/2026/Conference — ICLR 2026 Conference Withdrawn Submission_

### Official Review · Reviewer_ovtK · 2025-10-16

**Soundness:** 2
**Presentation:** 2
**Contribution:** 2
**Rating:** 2
**Confidence:** 4

**Summary:**

This work introduces the Schrödinger Bridge with Doob’s h-Transform (ScooBDoob), a discrete bridge matching framework capable of generating optimal stochastic paths between initial and target ensembles. The model is first trained to learn discrete bridges over microstates with the empirical MSM transition matrix as reference, and leverages the Doob's h-Transform to target the terminal states. Moreover, a well-defined potential based on the empirical transition density is applied to encourage the likelihood of transitioning into sparsely sampled intermediate states. Experiments are conducted on both the MB potential and the AiB9 peptide, showing good performance for transition path sampling.

**Strengths:**

- This work presents originality in both problem formulation and algorithmic design. By integrating Doob's h-Transform into the Schrödinger bridge framework, it enables transition path sampling toward a specified target distribution. Moreover, the incorporation of a density-aware potential enhances the model’s ability to access rare states.

**Weaknesses:**

- The training still relies on empirical Markov State Models (MSMs), requiring pre-existing MD trajectories as training samples for each system. This dependence limits its out-of-distribution generalization capability, thereby greatly constraining the model’s practical applicability.
- The experiments, conducted on only two systems, are insufficient to adequately support the authors' claims. Furthermore, the paper lacks appropriate baseline methods for comparison, making the reported results difficult to interpret in a meaningful context.
- The organization of the experimental section is not entirely coherent. A large portion of the experimental results is placed in the appendix. In addition, several terms, such as dynamic timesteps, are not clearly defined, which may hinder reader comprehension.

**Questions:**

1. In Section 3.1, the authors claim that ScooBDoob addresses two major challenges in MD simulations: the need for well-defined force fields and the long simulation times required to capture rare events. However, based on the subsequent model design and experiments, the training process still relies on MSMs derived from MD trajectories via TICA, and a separate model must be trained for each system. This severely limits the model’s out-of-distribution generalization capability. The authors are encouraged to provide a reasonable explanation for this inconsistency.
2. The experimental section lacks comparable baselines, which makes the reported results difficult to interpret. For example, the results in Table 1 are of limited significance, as readers have no reference to determine what range of metric values should be considered acceptable. It is recommended that the authors include appropriate baselines and provide additional experimental results.
3. A substantial portion of the experimental content is placed in the appendix, which significantly reduces the paper’s readability. Moreover, several terms lack precise definitions, undermining the overall rigor of the work: for instance, the procedure for selecting dynamic timesteps is not sufficiently explained. The authors are advised to reorganize the structure of the experimental section to improve clarity.
4. In Algorithm 1, the potential $V(i)$ defined in line 167 is inconsistent with Equation (10). A reasonable explanation for this discrepancy should be provided.
5. ScooBDoob introduces a density-aware regularization term to encourage exploration of sparse intermediate states, which is theoretically sound. However, I would like to see ablation study results to empirically validate the effectiveness of this component.
6. The model is trained with a complex set of training objectives. I believe it is necessary to conduct ablation studies to verify the contribution and necessity of each objective.
7. The paper states that the trained model demonstrates zero-shot generalization across different temperatures on the Aib9 peptide system. To my understanding, the model is trained solely on MD trajectories at 400 K, meaning that the learned transition probabilities correspond exclusively to the 400 K distribution. How does the model achieve zero-shot generalization to other temperatures? I would appreciate clarification from the authors on this point.
8. There is a typo in line 86, where it should be $P_0=\mu_0$. In addition, the symbol $T$ has not been defined earlier in the text.

**Details Of Ethics Concerns:**

None.

---

### Official Review · Reviewer_kNgk · 2025-10-26

**Soundness:** 1
**Presentation:** 2
**Contribution:** 1
**Rating:** 2
**Confidence:** 5

**Summary:**

This paper introduces ScooBDoob, a framework that combines Schrödinger Bridge objectives with Doob’s h-transform to construct transition matrices/generators that produce time-conditioned dynamics between prescribed endpoint ensembles. The mathematical presentation is compact and the approach is technically sound: the transform gives the correct conditioned generator and the authors’ numerical examples demonstrate that the method can produce endpoint-conditioned rollouts.

**Strengths:**

- The manuscript connects two well-known probabilistic tools (Schrödinger Bridges and Doob transforms) in a way that is natural for constructing endpoint-conditioned Markov dynamics.
- The method is theoretically principled: it produces conditional dynamics consistent with the original generator rather than introducing arbitrary bias.
- Experiments on Müller–Brown and Aib9 show the approach can be implemented and produce plausible conditioned trajectories.

**Weaknesses:**

- Core motivation. The authors explain that "eigendecomposition of transition matrices is a crucial step for extracting dynamical information" which often correct, and then highlight the gap "but can lead to numerical instability and inaccuracies if eigenvectors are unconstrained." In practice these are rarely limitations, as most cases often use estimators that explicitly enforce microscopic reversibility which avoids these problems [see work from Frank Noe's group from about 10 years ago, there are both maximum likelihood and Bayesian estimators, and estimators -- others have presented similar methods e.g. Bowman, Pande etc.]. There are several examples in the literature where MSM-style models of enormous systems have successfully been estimated.


- Practical novelty vs existing MSM/TPT methods. The manuscript does not convincingly demonstrate a practical advantage over established MSM-based TP analysis and committor analysis approaches (e.g., Szabo, Metzner, [1,2] and related work). From the viewpoint of molecular kinetics, many of the objects and observables targeted by ScooBDoob are already available via standard analyses, implemented in broadly accessible softwares such as deeptime, msmbuilder and pyemma. The authors must (a) clearly explain what ScooBDoob adds beyond these methods, and (b) include direct comparisons (baselines) to committor/TPT-derived path ensembles 1) https://pmc.ncbi.nlm.nih.gov/articles/PMC6910584/ 2) https://epubs.siam.org/doi/10.1137/070699500


- Misattributions and literature coverage. The paper omits several directly relevant references on committor and transition-path analysis for Markov models (see e.g. the two references above). This gap gives the impression that the authors did not fully situate their contribution within the established MD/statistical-physics literature; several claims and attributions in the text should be checked against the cited works. Some misattributions include: In the section describing "data-centric approaches to MD" there are a random assortment of unrepresentative citations, while several pioneering references are missing:
a) https://proceedings.neurips.cc/paper_files/paper/2023/hash/a598c367280f9054434fdcc227ce4d38-Abstract-Conference.html
b)https://proceedings.neurips.cc/paper_files/paper/2023/hash/7274ed909a312d4d869cc328ad1c5f04-Abstract-Conference.html
c) https://pubs.acs.org/doi/abs/10.1021/acs.jctc.1c00809
d) https://www.science.org/doi/10.1126/science.aaw1147

In appendix A: Charron et al and Majewski et al are referred to as generative models (incorrect, both papers describe ML coarse-grained forcefields). Also Lewis 2025b is referred to as an energy-based model.

I want to stress that I have not carefully gone through all the attributions, just the references that I am familiar with, and I find this alarming.


- The experimental suite is limited to a toy potential and a short peptide. To claim broad applicability to biomolecular dynamics, the manuscript needs stronger demonstrations. Standard MSM based analysis are routinely performed on large-scale systems, including protein-protein association of [i,ii,iii]
	i) https://www.nature.com/articles/nchem.2785
	ii) https://www.nature.com/articles/s41467-022-31374-5
	iii)  https://www.pnas.org/doi/abs/10.1073/pnas.2313360121

- The notation is sometimes informal or inconsistent (mixing generators and transition matrices, reusing symbols). There are also possibly incorrect index usages in Eqns. (11-12) that must be fixed. These issues hinder reproducibility.

- The limitations are not clearly laid out. One striking omission is that the outlined approach inherits all the limitations of building MSMs, in the context that the authors claim to be particularly helpful. In practice, building an MSM typically requires years of dedicated experts who iteratively collect data and refine model to ensure convergence of several diagnostics. This whole process is a foundation for what this approach.


Minor comments:
Replace 'Dirac delta' with 'Kronecker delta' for discrete state spaces.

Be consistent with notation: avoid using Q to mean both generator and transition matrix; distinguish Q (generator) and P(t)=exp(tQ) (transition kernel) where appropriate.

Clarify whether h is time dependent (finite-T) or time-independent (harmonic) in each use-case; make explicit the distinction between finite-horizon SB conditioning and infinite horizon/committor conditioning.

V is used both as a regularizer and in an SVD decomposition - pick distinct letters.

Check index conventions in Eqns. 11-12 for typos/misplaced indices; ensure conservation of row sums for generators.

Unless, the key problems are resolved in a satisfactory way, e.g. the practical motivation over existing methods, i will maintain my score.

**Questions:**

See weaknesses above.

---

### Official Review · Reviewer_qipD · 2025-10-31

**Soundness:** 3
**Presentation:** 3
**Contribution:** 2
**Rating:** 2
**Confidence:** 3

**Summary:**

In this work the authors present a new method to learn the transition matrix given an end point after N time steps. To do so, they use molecular dynamics and the theory of markov state models (MSM). The transition matrix is learned after computing the “teacher” one using a loss based on Schrodinger bridges, Chapman Kolmogorov test, and Stiefel consistency restrain. They then sketch that the learned transition matrices might be used for “sampling” transitions at different temperatures. The advantage of this method is however very unclear with respect to the existing ones.

**Strengths:**

Some aspects are well explained

**Weaknesses:**

In the abstract the authors make claims that are not supported by experiments “ScooBDoob preserves spectral stability of slow modes during training, recovers rare transtions pathways […] and generalizes to unseen temperature”

1) The obtained transition pathways seem very rough and do not follow the minimum free energy path in the Muller Brown experiment. They probably give an idea of what is happening, but it is definitely not what would be obtained from molecular dynamics.
2) The generalization over unseen temperature is not very well explained. The population of the macrostates might also change with temperature, is it something that the authors see with the learned matrix?

3) In the Problem Setup section, the authors state that their methods bypasses the need for a force-field. This is a wrong statement, as their MSM depends on a molecular dynamics trajectory and, therefore, on a forcefield. This statement is repeated also in the conclusion, but it is wrong. I think the first paragraph of this section can be removed as it is not relevant here.

This would leave some space for the second weakness of this paper: TICA is mentioned, as well as the eigenvalues of the transition matrix. I think it would be better if the authors made a link between their work and the theory of the transfer operator. Moreover, for a conference such as ICLR, it would be nice to see some representation learning in the paper, instead of just using the linear TICA. How do the results change, if, instead of using TICA, one uses VAMPnets for example? Or the representation learning based on the transfer operator or its associated infinitesimal generator (https://arxiv.org/abs/2307.09912 and https://arxiv.org/abs/2406.09028)

**Questions:**

1) Is this complicated loss really necessary, what happens, when for example, you learn the matrices with a mean-square error loss?
2) In the Muller-Brown experiment, the authors use “committor-based biasing to enable transition paths to cross saddle points”. Could you please elaborate on this? It seems to me that you used some kind of biasing scheme to accelerate the dynamics. If this is the case, then the learned MSM will be the one of the perturbed dynamics
3) In the presentation of TICA, the authors present everything in terms of atomic positions, which are not invariant under global translation/rotation. So features must be used, we only see it in the appendix for the Aib9 protein. It would be nice to understand how this featurization is performed a bit earlier than in the appendix.

4) Could the authors compare to at least one other method, for example this one https://arxiv.org/abs/2410.07974

---

### Official Review · Reviewer_efqJ · 2025-10-31

**Soundness:** 2
**Presentation:** 1
**Contribution:** 2
**Rating:** 4
**Confidence:** 2

**Summary:**

This paper introduces ScoobDoob, fusing Schrodinger Bridge and Doob’s h-transform to meta-stable dynamics. The proposed method is validated on 2 dimension muller brown potential and a Aib9 peptide, to show that it preserves spectral stability of slow modes, recovers rare events such as transition pathways, and generalize across zero-shot between temperatures.

**Strengths:**

1. Method (originality)

The author fuse the Schrodinger brdige and Doob’s h-transform for an endpoint conditioning on Markov state models, addressing rare-event sampling.

**Weaknesses:**

1. No baselines or methods comparison (significance)

While one of the motivation of this work was enhancing robustness, stability, and generalization capabilities of MSM-based methods (introduction section and key advancements), there are no baselines to compare with the proposed method.

2. Inconsistent terminology and missing definitions (clarity, quality)

Some parts of the papers were written hard to understand. For example, the $V(i)$ definition scaling in algorithm 1 and equation 10 are opposite. The teacher and student concept is barely described in the method section and suddenly appears in the experiments section.

3. Transition pathway experiments (clarity)

There seems to be no results to evaluate the quality of the transition pathways, compared to prior works considering transition path sampling such as transition state energy, distance to target state, mentioned in the introduction section (row 51).

Minor

- row 81 \mu_{n} → \mu_{N}
- row 86 maybe \mathbb{P}=\mu_{0} and \mathbb{P}_{T}=\mu_{T} → \mathbb{P}_{0}=\mu_{0} and \mathbb{P}_{N}=\mu_{N}
- row 148 U\in\mathbb{R}^{n\times k} → U\in\mathbb{R}^{d\times k}
- row 233 $V(i)$ on Algorithm 1 and 4th row, equation 10
- row 294 z_\theta ?
- row 323 missing period at end of sentence
- row 414 Figure 3 → Figure 2
- row 958 Figure a1, no legend on the color

**Questions:**

1. Generator for each state $i$

The algorithm 1 states that the generator is trained for each state $i$. Does this refer to that the authors are using $\vert i\vert$ neural network, or doing training over each state $i$? Since the later methodology section seems to describe a network training for the full matrix

2. Loss ablation (equation 23)

The loss is composed of three terms, MSM, bridge, and steif. While I got a glimpse on why these objectives are needed, are there any ablation regarding the $\gamma$ value? Does these auxiliary term actually have effect?

3. Teacher and student

Which part of the method is actually the teacher and student? Is the teacher fixed from the data, or optimized with the student together?

---

### Note · Authors · 2025-11-28

**Comment:**

We would like to sincerely thank the Area Chairs and all reviewers for their time, detailed feedback, and thoughtful engagement with our paper, *ScooBDoob: Schrödinger Bridge with Doob’s h-Transform for Molecular Dynamics*. We deeply appreciate the constructive comments that helped us identify key areas for improvement in both clarity and scope.

For future versions, we plan to substantially expand the comparative and experimental analyses to strengthen the paper’s empirical foundation. Specifically, we will (1) include direct baselines against standard MSM and transition path sampling methods (i.e., committor and TPT analyses) to better position ScooBDoob’s practical value, (2) provide quantitative metrics for transition pathway quality beyond visualization, such as transition state energies and distances to target conformations, and (3) include ablations for each loss term and density-aware regularization to empirically validate their effects.

We also acknowledge the need for improved presentation. We'll ensure that the revised manuscript will clarify terminology (teacher–student distinction, potential definitions, dynamic timestep procedure), align notation across equations and algorithms, and explicitly derive the discrete Schrödinger Bridge objective that leads to the Doob-tilted kernel. Additionally, we will better position ScooBDoob within the broader MSM and transfer-operator literature, and correct several citation and attribution inconsistencies as noted by the reviewers.

Overall, we are very grateful for the constructive feedback and believe these revisions will lead to a substantially stronger and clearer presentation of ScooBDoob in future submissions.

With much appreciation,
The authors

**Withdrawal Confirmation:**

I have read and agree with the venue's withdrawal policy on behalf of myself and my co-authors.